# ⊘*MCP-Universe*: Benchmarking Large Language Models with Real-World Model Context Protocol Servers

## Abstract

The Model Context Protocol (MCP) has emerged as a transformative standard for connecting large language models (LLMs) to external data sources and tools, and it is rapidly gaining adoption across major AI platforms. However, existing benchmarks are overly simplistic and fail to capture real-world application challenges such as long-horizon reasoning and large, unfamiliar tool spaces. To address this critical gap, we introduce **MCP-Universe**, the first comprehensive benchmark specifically designed to evaluate LLMs on realistic and difficult tasks through interaction with real-world MCP servers. Our benchmark spans 6 core domains and 11 different MCP servers: *Location Navigation*, *Repository Management*, *Financial Analysis*, *3D Design*, *Browser Automation*, and *Web Searching*. To ensure rigorous evaluation, we carefully design execution-based evaluators, including format evaluators for compliance, static evaluators for time-invariant content matching, and dynamic evaluators that automatically obtain real-time ground truth for temporally sensitive tasks. Through extensive evaluation of more than 20 leading LLMs, we find that even frontier models such as GPT-5-High (44.16% success rate) and Grok-4 (33.33% success rate) exhibit significant performance limitations. In addition, our benchmark poses a substantial long-context challenge, as the number of input tokens increases rapidly with each additional interaction step. It also introduces an unknown-tools challenge, since LLM agents often lack familiarity with the precise usage of certain MCP servers. Notably, enterprise-level agents such as Cursor and Claude Code fail to achieve better performance than the ReAct framework. Beyond evaluation, we open-source our extensible evaluation framework, enabling seamless integration of new LLMs, agents and MCP servers.

## 1 Introduction

The Model Context Protocol (MCP), introduced by Anthropic (2024a), represents a major paradigm shift in how AI systems interface with external data sources and tools. Dubbed the "USB-C of AI" (Rick, 2025), MCP addresses the long-standing issue of fragmented, bespoke integrations that trap language models in isolated information silos (Edwin, 2025). Since its release, MCP has gained rapid traction: major AI providers, including OpenAI (2025a) and Google (2025), have committed to adoption, while development platforms such as Cursor (2025) and Cline (2025) have begun integrating it. Despite its transformative potential, current evaluations remain insufficient, as existing benchmarks focus on narrow aspects of LLM performance, such as math reasoning (Cobbe et al., 2021) or function calling (Patil et al., 2025), without assessing how models interact with real-world MCP servers. MCP-RADAR (Gao et al., 2025) adapts datasets like HumanEval (Chen et al., 2021) and GSM8k (Cobbe et al., 2021) to the MCP setting, but these remain largely derivative and fail to capture the breadth of real-world applications or mitigate issues like data leakage Balloccu et al. (2024). Similarly, MCPWorld (Yan et al., 2025) continues to rely heavily on graphical user interfaces (GUIs) and provides limited coverage of MCP-enabled scenarios, restricting its utility for evaluating LLMs in authentic MCP-driven environments.

To address these critical limitations, we introduce our **MCP-Universe**, a benchmark aiming at evaluating LLMs in realistic, challenging use cases with real-world MCP servers. As shown in Figure 1, MCP-Universe captures realistic challenges: real-world tools usage, long-horizon multi-

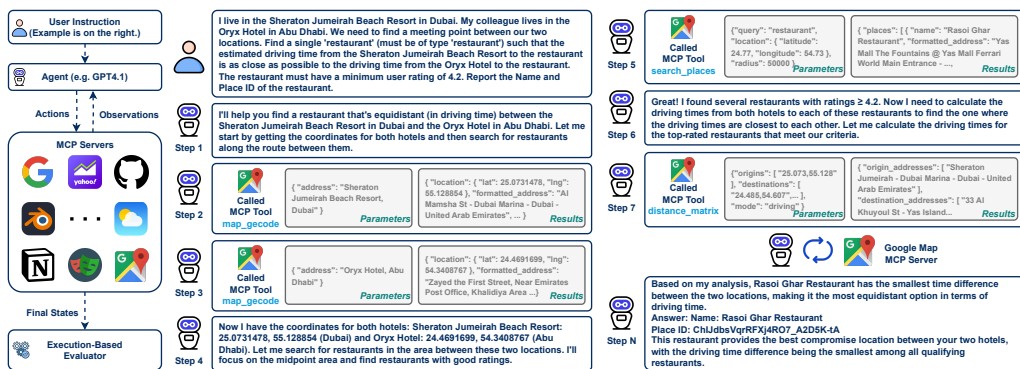

Figure 1: Example from MCP-Universe illustrating realistic challenges, including real-world tool usage, long-horizon multi-turn tool calls, long context, scattered evidence, and large tool spaces, with all challenges grounded in real-world MCP servers connected to actual environments.

turn tool calls, long context windows, scattered evidence, and large tool spaces. Unlike existing works, MCP-Universe is grounded in real-world MCP servers that connect to actual data sources and environments. Our benchmark encompasses 6 core domains, with 11 MCP servers spanning diverse applications: *Location Navigation*, *Repository Management*, *Financial Analysis*, *3D Design*, *Browser Automation*, and *Web Searching*, comprising a total of 231 tasks. Each domain captures the operational complexities of real-world deployments, from handling authentic financial data and navigating complex geospatial information to managing version control workflows and executing real-time ticket price checks.

To ensure rigorous evaluation, we carefully design execution-based evaluators rather than relying on LLM-as-a-judge (Zheng et al., 2023) (e.g., MCPEval (Liu et al., 2025) and LiveMCPBench (Mo et al., 2025)), recognizing that many tasks involve real-time data that static LLM knowledge cannot properly assess. Our evaluation includes format evaluators for agent format compliance, static evaluators for time-invariant content matching, and dynamic evaluators that automatically obtain real-time ground truth for temporally sensitive tasks. Furthermore, for the evolving nature of MCP servers, we provide an extensible, user-friendly framework that enables researchers and the broader community to seamlessly integrate new agents and MCP servers into the evaluation pipeline.

We conducted extensive experiments using MCP-Universe across all 6 core domains and 11 different MCP servers. Through extensive evaluation of more than 20 leading LLMs, we find that even top-performing models such as GPT-5-High (44.16% success rate), Grok-4 (33.33% success rate), and Claude-4.0-Sonnet (29.44% success rate) exhibit significant performance limitations, revealing a substantial gap between their impressive general capabilities and their effectiveness in real-world MCP environments. Our comprehensive analysis identifies several fundamental challenges that current LLM agents face in MCP interactions. First, we observe a *long-context challenge*, as the number of tokens increases rapidly with the number of interaction steps, often leading to context overflow and degraded performance in multi-step tasks. Second, there exists an *unknown-tools challenge*, where LLM agents often lack familiarity with the precise usage patterns, parameter specifications, and expected behaviors of diverse MCP servers. Additionally, our evaluation reveals significant *cross-domain performance variations*, with models showing markedly different success rates across different application domains, suggesting domain-specific optimization needs. Notably, enterprise-level agents like Cursor and Claude Code cannot achieve better performance than standard ReAct frameworks, highlighting the challenges of our benchmark.

In summary, this work makes the following key contributions:

- We introduce **MCP-Universe**, the first comprehensive benchmark for LLMs in MCP environments across 6 domains with real-world servers, where even SOTA LLMs struggle.

- We develop a rigorous **execution-based** evaluation framework with format, static, and dynamic assessment capabilities to enable comprehensive performance measurement.

- We reveal fundamental limitations of current LLM agents, such as challenges with long contexts, handling unknown tools, and cross-domain discrepancies, thereby highlighting directions for future MCP-agent design.

## 2 RELATED WORK

**MCP and LLMs as Agents.** MCP, introduced by Anthropic in late 2024, is an open standard for integrating AI with external data sources and tools (Anthropic, 2024a). Using a universal JSON-RPC 2.0 interface over STDIO and SSE (Edwin, 2025), it addresses the "data silo problem" and connects hosts (AI applications), clients (bridges), and servers (capability providers). Meanwhile, LLMs have progressed from text generators to autonomous agents with planning, reasoning, and tool use abilities (Wang et al., 2024a), enabled by advances in instruction following (Hida et al., 2024; Dong et al., 2025; Qian et al., 2025; Pyatkin et al., 2025), multi-step reasoning (Xu et al., 2025; Wei et al., 2022; Yao et al., 2023a; Zhang et al., 2025b), and tool integration (Qu et al., 2025; Qin et al., 2024b; Wang et al., 2024b; Zhang et al., 2025a). Agent paradigms such as ReAct (Yao et al., 2023b), Reflection (Shinn et al., 2023), and Plan-and-Solve (Wang et al., 2023), along with frameworks like AutoGen (Wu et al., 2023), MetaGPT (Hong et al., 2024), Camel-AI (Li et al., 2023a), and Lang-Graph (LangChain, 2024), demonstrate practical implementations. With multimodal LLMs (Hurst et al., 2024; Anil et al., 2023), GUI-based computer-use agents (Hu et al., 2024; Yang et al., 2025c; Li et al., 2025b; Yang et al., 2025b) have emerged, exemplified by OpenAI's CUA (OpenAI, 2025b), Anthropic's Computer-Use (Anthropic, 2024b), and ByteDance's UI-Taris (Qin et al., 2025), opening new frontiers in computer automation.

**Evaluation of Agents.** Evaluating LLM-based agents has become a major research area, with benchmarks targeting different aspects of agent capability. Web navigation has been widely studied through environments such as Mini-Wob++ (Liu et al., 2018), Mind2Web (1 & 2) (Deng et al., 2023; Gou et al., 2025), We-bLINX (Lù et al., 2024), AssistantBench (Yoran

Table 1: Contemporary MCP Benchmarks.

| Benchmark | Real-World Integration | Temporal Dynamics | Execution Evaluation |
|---|---|---|---|
| MCPWorld | ✓ | ✗ | ✓ |
| MCP-RADAR | ✗ | ✗ | ✓ |
| MCPEval | ✗ | ✓ | ✗ |
| LiveMCPBench | ✓ | ✓ | ✗ |
| **MCP-Universe** | ✓ | ✓ | ✓ |

et al., 2024), WebArena (Zhou et al., 2024), VisualWebArena, and VideoWebArena (Koh et al., 2024). GUI-based interaction is tested by OSWorld (Xie et al., 2024), WindowsAgentArena (Bonatti et al., 2024), and UI-Vision (Nayak et al., 2025), while software engineering tasks are covered by SWE-bench (Jimenez et al., 2024) and DevBench (Li et al., 2024). Tool and function calling has also been emphasized, with APIBank (Li et al., 2023b), ToolBench (Qin et al., 2024a), GAIA (Mialon et al., 2024), AppWorld (Trivedi et al., 2024), $\tau$-Bench (Yao et al., 2024), and BFCLv3 (Patil et al., 2025) evaluating agents' proficiency in invoking external APIs.

Recently, several MCP-specific benchmarks have been introduced (Table 1). MCPWorld (Yan et al., 2025) evaluates agents in GUIs and MCP environments but relies heavily on GUIs and omits time-varying tasks. MCP-RADAR (Gao et al., 2025) adapts datasets like HumanEval and GSM8k, but its tasks lack real-world grounding and temporal variation. MCPEval (Liu et al., 2025) and LiveMCP-Bench (Mo et al., 2025) both adopt LLM-as-a-Judge, which is ill-suited for real-time tasks and prone to style bias (Li et al., 2025a). In contrast, MCP-Universe integrates authentic MCP servers, temporal dynamics, and execution-based evaluation, providing a comprehensive benchmark that directly measures task completion in real-world MCP scenarios.

## 3 MCP-UNIVERSE

### 3.1 OVERVIEW

**MCP-Universe** is a comprehensive evaluation framework designed to assess the capabilities of LLMs when interacting with real-world MCP servers for challenging and practical tasks. As shown in Figure 2, our benchmark encompasses three core components: (1) an extensible, easy-to-use evaluation framework; (2) a collection of carefully designed task instructions grounded in real-world MCP server scenarios; (3) a suite of execution-based evaluators for measuring task completion.

To formalize the setting, we model the benchmark as follows. Let $S = \{s_1, s_2, \ldots, s_k\}$ denote the collection of MCP servers, where each server $s_i$ exposes a set of tools $T_i = \{t_{i,1}, t_{i,2}, \ldots, t_{i,|T_i|}\}$ through the MCP protocol. A task $\tau$ is defined as a tuple $(G, C, T_{\text{available}})$, where:

- $G$ is the goal specification describing the desired outcome;

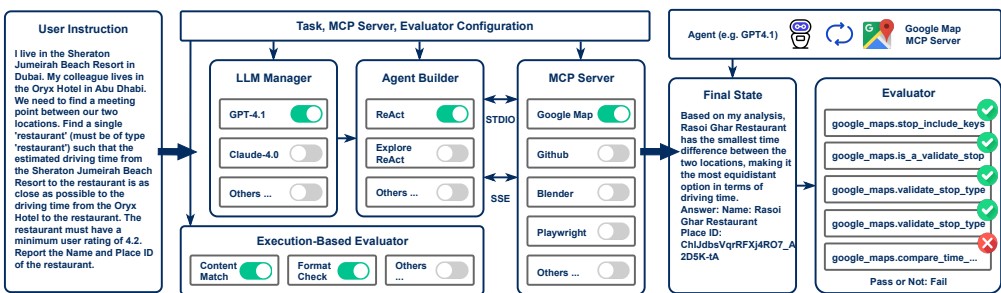

Figure 2: Overview of the MCP-Universe evaluation framework, which automatically configures LLM agents, MCP servers, and evaluators per task. Each evaluation consists of agent–server interactions via MCP, followed by automated execution-based assessment.

- $C$ contains the initial context and any relevant background information;

- $T_{\text{available}} = \bigcup_{i \in I} T_i$ is the set of tools accessible for the task, with $I \subseteq \{1, \ldots, k\}$ indicating which servers are available.

The benchmark challenges an agent to identify, sequence, and invoke appropriate tools from $T_{\text{available}}$ to achieve $G$ given $C$, requiring reasoning over partial information, adapting to diverse tool interfaces, and handling ambiguities or failures in tool responses.

For evaluation, let $M = \{m_1, m_2, \ldots, m_n\}$ be the set of language models and $A = \{a_1, a_2, \ldots, a_p\}$ be the set of agent patterns (e.g., ReAct) that can be paired with them. For a given $(m, a) \in M \times A$ and task $\tau \in \mathcal{T}$, the interaction produces a conversation trace $R = (r_1, r_2, \ldots, r_T)$, where each $r_t$ contains the agent's output and any tool invocations. The evaluation function

$$E : M \times A \times \mathcal{T} \to \{0, 1\}$$

assigns 1 if the task is successfully completed according to predefined success criteria, and 0 otherwise. Success is determined through a combination of automated checks (e.g., verifying structured outputs or end states). Aggregating $E(m, a, \tau)$ over all tasks yields a quantitative measure of an agent's proficiency in MCP-driven tool use.

## 3.2 EVALUATION FRAMEWORK

As illustrated in Figure 2, our evaluation framework coordinates multiple components to deliver objective and reproducible evaluation. Given a task specification, the framework automatically configures the entire evaluation pipeline: it builds the appropriate combination of LLM and agent, selects the required MCP servers, and sets up the corresponding evaluators. This configuration also manages resources, API endpoints, and evaluation criteria to ensure systematic execution. The LLM Manager supports multiple SOTA models, including GPT-5 and Grok-4, and is responsible for model configuration, API management, and standardized prompt formatting so that comparisons across models are fair and consistent. The Agent Builder constructs specialized agents such as ReAct and Function-Call and equips them with reasoning strategies tailored for MCP communication, enabling controlled comparison of different agent architectures within a uniform framework.

The framework further integrates with a wide range of MCP servers that expose real-world APIs, authentication mechanisms, and tool specifications. Through dynamic configuration, it supports both single server and multi server evaluation scenarios, ensuring that tasks reflect authentic operating environments rather than simplified simulations. To verify outcomes, the framework employs execution-based evaluators that apply domain-specific validation strategies, for example, stop type validation for Google Maps and branch checking for GitHub. This approach produces binary pass or fail results without relying on subjective judgments or expensive human annotation. Each evaluation proceeds through agent and server interactions mediated by MCP, followed by automated assessment, while detailed interaction logs are captured to provide comprehensive insights into model behavior across different tasks, servers, and evaluation settings.

Figure 3: Distribution of tasks in MCP-Universe across different application domains.

Table 2: Key statistics in MCP-Universe.

| Statistic | Number |
|---|---|
| Total tasks | 231 (100%) |
| - Location Navigation | 45 (19.5%) |
| - Web Searching | 55 (23.8%) |
| - Browser Automation | 39 (16.9%) |
| - 3D Designing | 19 (8.2%) |
| - Financial Analysis | 40 (17.3%) |
| - Repo. Management | 33 (14.3%) |
| Total MCP Servers | 11 |
| Total Tools in Servers | 133 |
| Total Unique Evaluators | 84 (100%) |
| - Format Evaluators | 4 (4.8%) |
| - Static Evaluators | 32 (38.1%) |
| - Dynamic Evaluators | 48 (57.1%) |

### 3.3 REAL-WORLD MCP SERVERS

A key principle of MCP-Universe is its reliance on real-world MCP servers rather than simulated environments, ensuring that evaluation reflects the authentic complexity of practical applications. As shown in Table 2, the benchmark spans 11 MCP servers with 133 tools across 6 core domains:

1. **Location Navigation**: Geographic reasoning and spatial task execution using the official Google Maps MCP server, which provides a rich suite of tools such as location search, route planning, and distance computation. Models must navigate the full complexity of real-world location data to complete navigation tasks effectively.

2. **Repository Management**: Repository development and codebase operations with the GitHub MCP server, exposing tools for repository search, issue tracking, and code editing. Models must engage with the authentic challenges of real-world repository management to execute development tasks effectively.

3. **Financial Analysis**: Quantitative reasoning in dynamic markets using the Yahoo Finance MCP server, which supports stock monitoring, shareholder lookup, and options tracking over live financial data. Models must analyze live and volatile financial information to make reliable decisions under uncertainty.

4. **3D Designing**: Professional computer-aided design tasks supported by the Blender MCP server, offering object creation, asset manipulation, and material setup. Models must operate within the technical depth of professional design workflows to generate valid outputs.

5. **Browser Automation**: Automated interaction with web applications via the Playwright MCP server, enabling browser navigation, button clicking, and page snapshotting. Models must handle the intricacies of real-world browser environments to perform web automation tasks.

6. **Web Searching**: Open-domain information seeking using the Google Search MCP server for queries and the Fetch MCP server for URL content retrieval. Models must sift through open-ended, noisy web data to identify and synthesize relevant information.

This selection emphasizes both domain diversity and real-world relevance. To further increase coverage and task complexity, additional MCP servers are included, such as Notion, Weather, Date, and Calculator. Details of all MCP servers are provided in Appendix A.

### 3.4 TASKS AND EVALUATORS

Since MCP is new and lacks high-quality usage examples, we manually designed challenging tasks to reflect real use cases. Tasks that can be trivially solved by LLMs without MCP servers, or consistently completed within five retries using MCP servers, are discarded and replaced. As shown in Figure 3, each domain contains 4-5 task types covering representative scenarios. For example, Location Navigation includes route planning, optimal stops, location search, and place finding; Repos-

Table 3: Evaluation with LLMs using ReAct. We report success rate (SR), average evaluator score (AE; mean percentage of evaluators passed), and average steps (AS) for successful tasks. †: GPT-OSS-120B does not follow the ReAct pattern, and is therefore evaluated with OpenAI's Agent SDK.

| Model | Location Navigation | Repository Management | Financial Analysis | 3D Designing | Browser Automation | Web Searching | Overall AE | AS | SR |
|---|---|---|---|---|---|---|---|---|---|
| *Proprietary Models* | | | | | | | | | |
| GPT-5-High | 26.67 | **30.30** | **67.50** | **57.89** | **43.59** | **45.45** | **62.82** | 6.84 | **44.16** |
| GPT-5-Medium | 33.33 | **30.30** | **67.50** | 52.63 | 35.90 | **45.45** | 60.23 | 8.22 | 43.72 |
| Grok-4 | 28.89 | 12.12 | 40.00 | 26.32 | 41.03 | 41.82 | 49.01 | 7.75 | 33.33 |
| Claude-4.1-Opus | 17.78 | 21.21 | 52.50 | 36.84 | 35.90 | 20.00 | 49.14 | 7.04 | 29.44 |
| Claude-4.0-Opus | 15.56 | 15.15 | 55.00 | 31.58 | 38.46 | 18.18 | 46.40 | 7.69 | 28.14 |
| Claude-4.0-Sonnet | 22.22 | 12.12 | 55.00 | 26.32 | 38.46 | 21.82 | 50.61 | 7.46 | 29.44 |
| Grok-4-Fast | 24.44 | 9.09 | 65.00 | 5.26 | 25.64 | 21.82 | 48.95 | 6.54 | 27.27 |
| Grok-Code-Fast-1 | 26.67 | 9.09 | 62.50 | 15.79 | 20.51 | 18.18 | 44.72 | 6.87 | 26.41 |
| o3 | 26.67 | 6.06 | 40.00 | 26.32 | 25.64 | 29.09 | 38.95 | 4.82 | 26.41 |
| o4-mini | 26.67 | 18.18 | 40.00 | 36.84 | 23.08 | 18.18 | 40.38 | 7.90 | 25.97 |
| Claude-3.7-Sonnet | 13.33 | 18.18 | 40.00 | 36.84 | 23.08 | 21.82 | 40.36 | 7.16 | 24.24 |
| Gemini-2.5-Pro | 13.33 | 12.12 | 50.00 | 21.05 | 25.64 | 12.73 | 36.93 | 6.98 | 22.08 |
| Gemini-2.5-Flash | 15.56 | 12.12 | 37.50 | 21.05 | 30.77 | 14.55 | 33.99 | 8.26 | 21.65 |
| GPT-4.1 | 8.89 | 6.06 | 40.00 | 26.32 | 23.08 | 10.91 | 41.32 | 5.24 | 18.18 |
| GPT-4o | 8.89 | 9.09 | 35.00 | 26.32 | 12.82 | 9.09 | 37.03 | 6.03 | 15.58 |
| *Open-Source Models* | | | | | | | | | |
| GLM-4.5 | 17.78 | 9.09 | 50.00 | 26.32 | 15.38 | 27.27 | 41.16 | 7.33 | 24.68 |
| Qwen3-Coder | 13.33 | 3.03 | 57.50 | 31.58 | 30.77 | 9.09 | 41.39 | 7.77 | 22.94 |
| DeepSeek-V3.1 | 15.56 | 0.00 | 42.50 | 31.58 | 28.21 | 18.18 | 43.23 | 6.31 | 22.08 |
| Kimi-K2 | 11.11 | 3.03 | 52.50 | 15.79 | 25.64 | 10.91 | 41.28 | 6.96 | 19.91 |
| GLM-4.5-Air | 17.78 | 6.06 | 42.50 | 10.53 | 17.95 | 16.36 | 37.42 | 6.42 | 19.48 |
| Qwen3-235B | 11.11 | 9.09 | 50.00 | 15.79 | 15.38 | 9.09 | 38.53 | 5.74 | 18.18 |
| DeepSeek-V3 | 11.11 | 6.06 | 30.00 | 26.32 | 12.82 | 7.27 | 35.82 | 5.06 | 14.29 |
| GPT-OSS-120B† | 6.67 | 6.06 | 35.00 | 10.53 | 5.13 | 5.45 | 26.34 | - | 11.26 |

itory Management includes project setup, issue tracking, automation setup, and code integration; Financial Analysis includes portfolio analysis, financial statements, trading strategies, institutional holdings, and dividend analysis; 3D Designing includes object creation, material setup, lighting, render settings, and scene hierarchy; Browser Automation includes travel booking, sports analytics, academic research, platform exploration, and map navigation; and Web Searching includes person identification, entity discovery, metric matching, complex reasoning, and factual lookup. All tasks are cross-checked by other authors for feasibility, clarity, and correctness.

To evaluate task completion, we design execution-based evaluators tailored to each task. While many recent works adopt the LLM-as-a-judge paradigm (Liu et al., 2025; Mo et al., 2025), we argue it is unsuitable for MCP-Universe because some tasks require real-time data, while LLM judges have static knowledge and are prone to bias or hallucination. Although creating execution-based evaluation requires more human effort, it is necessary for fairness and comprehensiveness. We design three evaluator types: (1) Format Evaluators check compliance with output requirements; (2) Static Evaluators assess correctness for tasks with fixed answers, such as city counts in route planning, historical stock prices, or football statistics; and (3) Dynamic Evaluators automatically obtain real-time ground truth for temporally sensitive tasks, such as flight prices, current weather, or the number of GitHub issues. Each evaluator is reviewed by the other authors for feasibility and accuracy. Appendix B provides more details and examples of both tasks and evaluators.

Based on these evaluators, we employ a complementary dual-metric system: **Average Evaluator Score** (AE) and **Success Rate** (SR). Following OSWorld (Xie et al., 2024), the AE represents the mean percentage of passed evaluators per task. This ensures models receive partial credit for progress, preventing a zero score due to a single missed evaluator. The SR serves as a strict binary metric to determine if the task was fully completed. These metrics work together to provide a balanced evaluation: the AE rewards progress, while the SR ensures that high partial scores do not mask the inability to fully complete the task.

## 4 EXPERIMENT

### 4.1 SETUP

In our experiments, we evaluate both proprietary and open-source SOTA LLMs on MCP-Universe. The models include xAI's Grok-4, Grok-4-Fast and Grok-Code-Fast-1 (xAI, 2025), An-

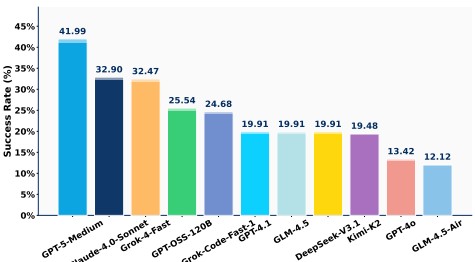 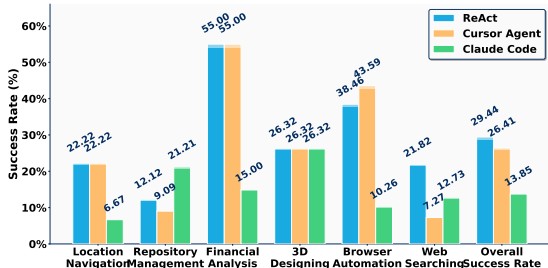

Figure 4: Evaluation of LLM performance with function calling (left) and enterprise-level agent frameworks (right).

thropic's Claude-4.1-Opus, Claude-4.0-Opus, Claude-4.0-Sonnet (Anthropic, 2025b), and Claude-3.7-Sonnet (Anthropic, 2025a), OpenAI's GPT-5 (OpenAI, 2025a), o3, o4-mini (OpenAI, 2025b), GPT-4.1 (OpenAI, 2025b), GPT-4o (Hurst et al., 2024), and GPT-OSS (OpenAI, 2025a), Google's Gemini-2.5-Pro and Gemini-2.5-Flash (Comanici et al., 2025), Zai's GLM-4.5 and GLM-4.5-Air (Zai, 2025), Moonshot's Kimi-K2 (Moonshot, 2025), Qwen's Qwen3-Coder and Qwen3-235B-A22B-Instruct-2507 (Yang et al., 2025a), and DeepSeek's DeepSeek-V3-0324 and DeepSeek-V3.1 (DeepSeek-AI et al., 2025). All models are top-ranked on the lmsys Chatbot Arena leaderboard (Chiang et al., 2024); open-source ones exceed 100B parameters. For agents, we adopt two settings: the widely used ReAct framework (Yao et al., 2023b), where models iteratively generate thoughts based on observations and then actions based on thoughts, and a native function calling mode, where models directly invoke MCP server functions without explicit ReAct style reasoning. In the latter, an LLM may call multiple tools within a single step. Additional setup details are provided in Appendix C.

## 4.2 MAIN RESULTS

**LLMs w/ ReAct.** As shown in Table 3, we compare SOTA proprietary and open-source LLMs on MCP-Universe. OpenAI's GPT-5 achieves the highest success rates, with high and medium reasoning effort reaching 44.16% and 43.72%, far ahead of others. Grok-4 ranks third (33.33%). GPT-5 leads across all domains, Location Navigation (33.33%), Repository Management (30.30%), Financial Analysis (67.50%), 3D Designing (57.89%), Browser Automation (43.59%), and Web Searching (45.45%), and also achieves the highest average evaluator score (62.82%). Grok-4 shows strength in Browser Automation (41.03%) and Web Searching (41.82%). Location Navigation remains difficult for all models (<35%), with GPT-4.1 and GPT-4o scoring below 10%. In Repository Management, only GPT-5 surpasses 30%, and in 3D Design only GPT-5 exceeds 50%. Among open-source models, GLM-4.5 performs best (24.68%), comparable to some proprietary models like o4-mini (25.97%) and Claude-3.7-Sonnet (24.24%), yet still 20 points behind the best.

Beyond success rates, we also assess average evaluator (AE) scores and average steps (AS). GPT-5 ranks highest in both SR and AE (62.82%), showing strong consistency. Discrepancies emerge: Claude-4.0-Sonnet achieves a slightly higher AE (50.61%) than Grok-4 (49.01%), but Grok-4 attains higher SR (33.33% vs. 29.44%). Similarly, Claude-4.1-Opus records 49.14% AE but only 29.44% SR. Task completion typically requires 5–8 steps: o3 averages 4.82 steps (26.41% SR), GPT-5 requires 6.84–8.22 depending on effort, and Grok-4 averages 7.75. These results highlight that even frontier LLMs remain unreliable across diverse real-world MCP tasks, underscoring MCP-Universe as a challenging and necessary testbed.

**LLMs w/ Funcation Call.** Beyond ReAct, we also evaluate representative LLMs using native function calling, where models directly invoke MCP server functions without explicit ReAct-style reasoning. In each step, an LLM can call more than one tool. As shown in Figure 4 (left), GPT-5-Medium achieves the highest success rate (41.99%), close to its ReAct counterpart (43.72%), while Claude-4.0-Sonnet performs slightly better with function calling (32.90% vs. 29.44%). Grok-4-Fast also benefits from function calling, improving from 27.27% with ReAct to 32.47%. GPT-4.1 and GPT-4o show only minor changes (19.91% vs. 18.18%, and 13.42% vs. 15.58%). Other LLMs generally lag behind, including Grok-Code-Fast-1 (24.68%), GLM-4.5 (19.91%), DeepSeek-V3.1 (19.91%), Kimi-K2 (19.48%), and GLM-4.5-Air (12.12%). Notably, GPT-OSS-120B reaches 25.54% under function calling, a dramatic improvement over its OpenAI Agent SDK

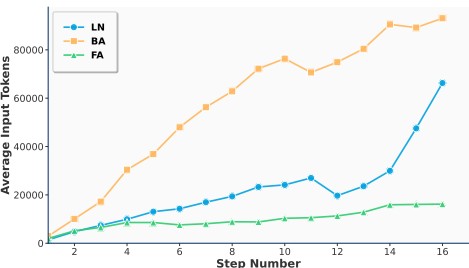 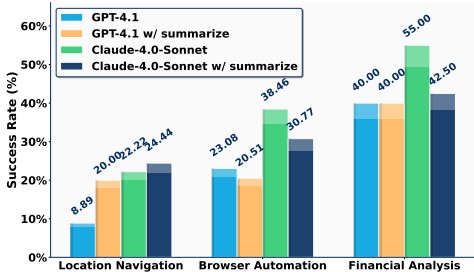

Figure 5: (Left) Growth of average context length (in tokens) as the number of interaction steps increases in MCP-Universe tasks, illustrating the long context challenge. (Right) Effect of introducing a summarization agent on LLM agent performance across selected domains.

result (11.26%), making it competitive with other open source models. This 14.28-point gain highlights strong sensitivity to interaction paradigms: some models, like GPT-OSS-120B, are more effective with direct function invocation than with ReAct. More results can be found in the Appendix D.

**Enterprise-Level Agents.** As shown in the right of Figure 4, we compare three agent configurations using Claude-4.0-Sonnet as the backbone: ReAct, Cursor Agent, and Claude Code. ReAct achieves the highest overall success rate (29.44%), followed by Cursor Agent (26.41%) and Claude Code (13.85%). Both Cursor Agent and Claude Code are primarily designed for code generation, explaining their weaker performance on general tasks. Notably, Claude Code achieves its best result in Repository Management (21.21%), outperforming ReAct (12.12%) and Cursor Agent (9.09%), showing its strength in code-related tasks. However, both struggle in general-purpose domains: Cursor Agent underperforms in Web Searching (7.27% vs. ReAct's 21.82%) due to reliance on internal tools, while Claude Code shows particularly poor results in Location Navigation (6.67%) and Financial Analysis (15.00%). These results highlight that specialized code agents, though effective in their domains, generalize poorly to broader real-world tasks.

## 4.3 ANALYSIS

**Evaluator-Specific Performance.** Our benchmark incorporates three evaluator types: format evaluators, static evaluators, and dynamic evaluators. Table 4 presents a breakdown of model performance across these types. For format evaluators, performance varies significantly across models: Claude-4.0-Sonnet achieves the highest success rate at 98.29%, followed by DeepSeek-V3.1 (96.58%), and GPT-4.1 (95.73%). However, some models struggle with format compliance, particularly Gemini-2.5-Flash (51.28%), Gemini-2.5-Pro (64.10%), and o3 (73.50%), indicating that format adherence is not universally mastered even among frontier models. We highlight the naive error in the Appendix E.6. On content-sensitive static evaluators, GPT-5-High leads with 69.59%, followed by Claude-4.0-Sonnet (61.92%), while most other models achieve around 40-50% suc-

Table 4: Success rate across different types of evaluators on our MCP-Universe benchmark.

| Model | Format | Static | Dynamic |
|---|---|---|---|
| GPT-5-High | 88.03 | **69.59** | 62.11 |
| GPT-5-Medium | 88.89 | 61.92 | **65.96** |
| Grok-4 | 88.03 | 49.04 | 52.98 |
| Claude-4.0-Sonnet | **98.29** | 61.92 | 54.74 |
| Grok-Code-Fast-1 | 85.47 | 52.60 | 50.53 |
| o3 | 73.50 | 38.63 | 43.16 |
| o4-mini | 78.63 | 44.66 | 43.86 |
| Gemini-2.5-Pro | 64.10 | 39.18 | 42.46 |
| Gemini-2.5-Flash | 51.28 | 45.21 | 30.88 |
| GPT-4.1 | 95.73 | 57.53 | 49.47 |
| GPT-4o | 91.45 | 54.79 | 45.61 |
| GLM-4.5 | 81.20 | 46.30 | 48.07 |
| GLM-4.5-Air | 87.18 | 42.47 | 47.02 |
| Kimi-K2 | 70.94 | 33.15 | 53.33 |
| Qwen3-Coder | 78.63 | 42.74 | 49.12 |
| Qwen3-235B | 92.31 | 43.29 | 53.68 |
| DeepSeek-V3.1 | 96.58 | 59.73 | 49.12 |

cess. For dynamic evaluators, GPT-5-Medium achieves the highest performance at 65.96%, followed by GPT-5-High (62.11%) and Claude-4.0-Sonnet (54.74%). The substantial performance gap between format evaluators (where top models exceed 90%) and content evaluators (where most models achieve 40-60%) indicates that the primary source of failure lies in content generation rather than format compliance. This demonstrates that our benchmark evaluates LLMs from multiple angles, including format compliance and content correctness under both static and dynamic conditions. Additional detailed performance analysis is provided in Appendix E.

**Long Context Challenges.** In our MCP-Universe benchmark, long context handling is a major challenge, especially in Location Navigation, Browser Automation, and Financial Analysis. These

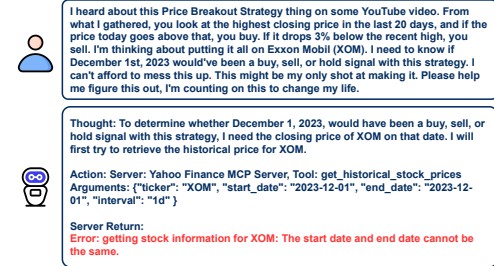
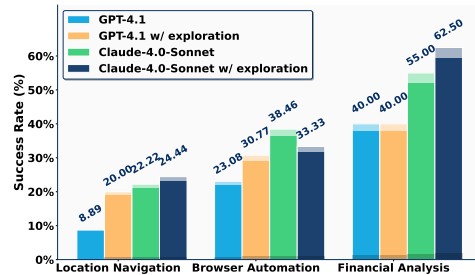

Figure 6: (Left) An example of the unknown tool challenges. (Right) Effect of introducing the exploration phase on LLM agent performance across selected domains.

tasks often require reasoning over extended sequences of observations or historical actions that exceed typical context windows. For instance, Google Maps servers may return detailed multi-location information, Playwright servers can output full webpage HTML, and Yahoo Finance servers provide stock data across long ranges, producing large volumes of contextual input. As shown in Figure 5, token counts grow rapidly with interaction steps, confirming long context as a core difficulty.[1] To explore mitigations, we tested a summarization agent that compresses server outputs at each step while attempting to preserve essential information (details in Appendix F). This method improved GPT-4.1 and Claude-4.0-Sonnet in Location Navigation but had little or negative effect in Browser Automation and Financial Analysis, where fine-grained details are critical. These results show that MCP-Universe introduces realistic long context challenges while exposing the limits of simple compression, underscoring its value as a testbed for advancing context handling methods in LLM agents.

**Unknown Tools Challenges.** In addition to long context challenges, our error analysis shows that LLMs often misuse MCP server tools due to limited familiarity with their interfaces. For example, as shown in Figure 6 (left), models frequently fail on Yahoo Finance by setting identical start and end dates, violating API requirements and causing execution errors. To mitigate this, we introduce an exploration phase where models freely interact with MCP tools to probe interfaces, learn parameter requirements, and build basic tool knowledge before moving to an exploitation phase that combines this knowledge with a ReAct-style framework to solve tasks (details in Appendix G). As shown in Figure 6 (right), exploration yields gains in some domains: GPT-4.1 improves by 7.69 points in Browser Automation (30.77%) and Claude-4.0-Sonnet by 7.50 points in Financial Analysis (62.50%), but results are mixed, with declines elsewhere. These outcomes highlight both the potential and the limitations of exploration: while it can enhance performance on tasks requiring tool familiarity and iterative reasoning, it is not universally effective, underscoring the need for more adaptive strategies and reinforcing MCP-Universe as a rigorous testbed for advanced LLM agents.

**More MCP Servers Connected.** In Table 3, each task is paired only with the MCP servers directly relevant to it. Here, we extend the setup by connecting additional, unrelated MCP servers to evaluate LLM performance under greater tool complexity. For all tasks, we connect seven MCP servers comprising 94 tools, which introduces noise and leads to a clear decline in performance (Figure 7). For instance, Claude-4.0-Sonnet's success rate in Location Navigation falls from 22.22% to 11.11%, GPT-4.1's Browser Automation accuracy drops from

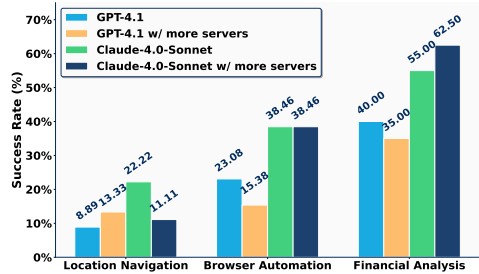

Figure 7: Effect of connecting with more unrelated MCP servers.

23.08% to 15.38%, and its Financial Analysis score decreases from 40.00% to 35.00%. These results show that MCP-Universe also functions as a robust testbed for assessing LLM resilience when faced with larger and less relevant toolsets.

---

[1]The context length experiment is based on Claude-4.0-Sonnet.

## 5 CONCLUSION

In this work, we introduce **MCP-Universe**, the first benchmark to rigorously evaluate LLMs in real-world MCP environments. By grounding tasks in authentic data and using execution-based evaluators, it reveals key gaps in long context handling, tool usage, and cross-domain performance. Our experiments show that even frontier models and enterprise-level agents struggle with these challenges, highlighting the need for advances in model design and agent integration. With its extensible framework, MCP-Universe offers a valuable testbed for driving progress toward more reliable real-world MCP-Use applications.

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

Table 5: Evaluation with LLMs using Function-Calling.

| Model | Location Navigation | Repository Management | Financial Analysis | 3D Designing | Browser Automation | Web Searching | Overall AE | Overall AS | Overall SR |
|---|---|---|---|---|---|---|---|---|---|
| *Proprietary Models* | | | | | | | | | |
| 🌀 GPT-5-Medium | 35.56 | 30.30 | 60.00 | 52.63 | 43.59 | 36.36 | 59.29 | 7.85 | **41.99** |
| 𝖠\ Claude-4.0-Sonnet | 22.22 | 6.06 | 77.50 | 36.84 | 35.90 | 21.82 | 52.61 | 9.78 | 32.90 |
| ⊘ Grok-4-Fast | 22.22 | 6.06 | 80.00 | 21.05 | 23.08 | 32.73 | 52.20 | 7.25 | 32.47 |
| ⊘ Grok-Code-Fast-1 | 17.78 | 12.12 | 57.50 | 26.32 | 15.38 | 20.00 | 43.89 | 7.61 | 24.68 |
| 🌀 GPT-4.1 | 15.56 | 6.06 | 45.00 | 26.32 | 28.21 | 5.45 | 39.58 | 6.83 | 19.91 |
| 🌀 GPT-4o | 8.89 | 6.06 | 35.00 | 10.53 | 12.82 | 7.27 | 32.43 | 6.65 | 13.42 |
| 🜸 Qwen3-Max-Preview | 13.33 | 15.15 | 40.00 | 21.05 | 15.38 | 5.45 | 36.71 | 8.57 | 17.32 |
| *Open-Source Models* | | | | | | | | | |
| 🌀 GPT-OSS-120B | 24.44 | 15.15 | 42.50 | 36.84 | 20.51 | 20.00 | 39.78 | 7.53 | **25.54** |
| ❋ GLM-4.5 | 22.22 | 9.09 | 37.50 | 26.32 | 10.26 | 16.36 | 37.58 | 10.17 | 19.91 |
| 🐳 DeepSeek-V3.1 | 11.11 | 6.06 | 45.00 | 26.32 | 23.08 | 12.73 | 38.88 | 10.74 | 19.91 |
| ℍ Kimi-K2 | 17.78 | 6.06 | 42.50 | 15.79 | 23.08 | 10.91 | 38.52 | 10.20 | 19.48 |
| ❋ GLM-4.5-Air | 8.89 | 3.03 | 25.00 | 10.53 | 7.69 | 14.55 | 26.47 | 12.21 | 12.12 |

Table 6: MCP servers in our benchmark and their links.

| MCP Server | URL |
|---|---|
| Google Map MCP | https://github.com/modelcontextprotocol/servers-archived/tree/main/src/google-maps |
| Github MCP | https://github.com/github/github-mcp-server |
| Yahoo Finance MCP | We will release this server upon acceptance. |
| Blender MCP | We will release this server upon acceptance. |
| Playwright MCP | https://github.com/microsoft/playwright-mcp |
| Google Search MCP | We will release this server upon acceptance. |
| Fetch MCP | https://github.com/modelcontextprotocol/servers/tree/main/src/fetch |
| Notion MCP | https://github.com/makenotion/notion-mcp-server |
| Weather MCP | We will release this server upon acceptance. |
| Date MCP | We will release this server upon acceptance. |
| Calculator MCP | https://pypi.org/project/mcp-server-calculator/ |

# A    MCP SERVERS

As shown in Table 6, we include the names and links of all MCP servers in our benchmark to help users more easily utilize the benchmark. Most of them are official MCP servers, and some are based on the official APIs to ensure the quality of the servers.

# B    TASKS AND EVALUATORS EXAMPLES

In Table 7, 8, 9, 10, 11, and 12, we include 30 task examples of our benchmark. In Table 13, 14, and 15, we include 3 examples of the evaluators of our benchmark. All tasks and evaluators will be released upon acceptance.

# C    SETUP

As shown in Table 16, we present the versions of the LLMs used in our evaluation. The temperature is set to 1.0 for all LLMs. In Figure 11, we present the ReAct prompt used in our experiments. In Figure 12, we present the function-call prompt used in our experiments.

# D    LLMS W/ FUNCTION-CALL PERFORMANCE

As shown in Table 5, we show the details performance of LLMs w/ Function-Call on our MCP-Universe.

Table 7: Examples of Location Navigation Tasks. We do not include the format requirements to save space.

**Example 1:** Hi! My partner and I are planning a special pre-wedding road trip from Los Angeles to San Francisco as one last adventure before we tie the knot! We want to make this journey memorable before we start our married life together. Our plan is to drive through exactly 3 interesting cities between the starting and ending points to really enjoy this time together. Could you please map out exactly 2 distinct driving route options for this pre-wedding celebration? Oh, we must visit friends in Coalinga during our trip to share our exciting news with them! We're so excited about this adventure before our big day!

**Example 2:** I need to drive from the Merlion Park, Singapore to the Petronas Towers, Kuala Lumpur, Malaysia. Please plan a driving route. Along this route, I need to make exactly one stop. Find the single location (report its name and Place ID) that is closest to the geographic midpoint of the calculated route (based on the route polyline) and is categorized as either a gas station OR a restaurant with a user rating of at least 4.2.

**Example 3:** My wife and I are planning an amazing family adventure from Disneyland in Anaheim to Yosemite Valley Visitor Center with our wonderful kids! As a devoted husband and father, I want to make sure everyone stays happy and comfortable during our journey, so I need help creating a perfect driving route with four thoughtfully chosen family-friendly stops. Could you please map out a route with exactly four intermediate points that are located at the geographic fifth points along the route (based on the route polyline)? For each stop, I'd love to find locations (please provide names and Place IDs) that are either a restaurant where we can all enjoy a meal together, a comfortable hotel where my family can rest, or a reliable gas station to keep our adventure going. All with a minimum user rating of 4.2 to ensure the best experience for my loved ones. This trip should be both practical and create wonderful memories for our entire family!

**Example 4:** I live in Kent Ridge Hill Residences, Singapore. One of my friends lives in Symphony Suites, Yishun, Singapore. Another friend lives in Katong Gardens, Singapore. We're looking for a cozy spot to catch up and chat! Can you help us find a meeting point between our 3 locations? We'd love to find a single cafe (must be of type 'cafe') where we can all gather comfortably, ideally somewhere where the estimated driving time from each of our places to the cafe is as close as possible. Please report the Name and Place ID of the cafe.

**Example 5:** Identify 1 library location in New York City that are north of the latitude of Queensbridge Park AND east of the longitude of NewYork-Presbyterian/Weill Cornell Medical Center.

Table 8: Examples of Repository Management Tasks.

**Example 1:** For this assignment, I would like you to establish a new project repository named `ai-code-reviewer`. Please begin by initializing the repository with three branches: `feature-analysis`, `feature-integration`, and `main`. You should include an initial `README.md` file in the main branch with the content "# AI Code Reviewer\n\nAn intelligent code review assistant that analyzes code quality, detects potential bugs, and suggests improvements using machine learning techniques.". Next, please add `code_analyzer.py` in the `feature-analysis` branch with the content "# Code analysis module\nimport ast\n\nclass CodeAnalyzer:\n def __init__(self, code):\n self.code = code\n self.tree = ast.parse(code)\n\n def analyze(self):\n # TODO: Implement analysis logic\n pass". Additionally, create a `.gitignore` file in the main branch with the exact content: "# Python cache and virtual environments\n__pycache__/\n*.pyc\n*.py.class\nvenv/\n*.env\n\n# Analysis results\nreports/\nlogs/\n\n# Model checkpoints\nmodels/". Please copy `train.py` from bigcode-project's starcoder repository to the `feature-integration` branch. Finally, I would like you to create a pull request to merge `feature-analysis` into `main` with the title "Add initial code analysis module" and description "This PR implements the basic code analysis module using AST parsing for initial code quality assessment."

**Example 2:** Hi! I'm learning how to use GitHub and I want to practice exploring repositories and working with issues. Can you help me with a research project? I'd like to search for repositories owned by 'google' that have 'generative-ai' in their name. Once I find them, I want to count how many open issues each repository has that are labeled 'type:bug'. This will help me understand how developers track bugs in real projects! After gathering this information, I need to practice creating my own repository called 'google-generative-ai-issues' and uploading a CSV file named 'google_generative_ai_bug_report.csv' to it. The CSV should have two columns: 'repository_name' and 'open_bug_count'. This exercise will help me learn about repository management, issue tracking, and data organization on GitHub!

**Example 3:** There are two repositories: QwenLM's Qwen2.5-VL and deepseek-ai's DeepSeek-VL2. Fork the repository with the fewest open issues, maintaining the same name as the source repository. If Qwen2.5-VL is forked, add a reference link at the bottom of the `README.md` file: 'Related project: [DeepSeek-VL2](the link of DeepSeek-VL2 repo)'. If DeepSeek-VL2 is forked, add a reference link at the bottom of the `README.md` file: 'Related project: [Qwen2.5-VL](the link of Qwen2.5-VL repo)'.

**Example 4:** Hi! I'm a student working on learning GitHub automation and I really need your help. Could you please help me create a new project repository named `auto-issue-close`? I need to initialize it with just the main branch and include an initial `README.md` file with the content "# Automated Issue Closing\n\nA repository to test GitHub automation for closing labeled issues." I'm struggling with GitHub automation workflows and would really appreciate your help developing a script that automatically closes issues labeled as 'completed' or 'wontfix'. After we set up the automation script, I need to test it by creating three sample issues with different labels (labels: 'completed', title: "Implement new feature"; labels: 'wontfix', title: "Remove legacy code"; labels: 'bug', title: "Fix login error"). I'm really grateful for any assistance you can provide!

**Example 5:** Hi! I need help with a research project. Could you please search for repositories owned by 'huggingface' with 'diffusers' in the name? For each repository you find, I'd like to know how many open issues are labeled with 'bug'. Then, could you help me create a CSV file called `diffusers_bug_report.csv` and put it in a new repository called `huggingface-diffusers-issues` under my account? If the repository doesn't exist yet, please create it for me. The CSV should have two columns: `repository_name` and `open_bug_count`, with each row showing the full repository name and how many open bug issues it has. Thanks so much for your help!

Table 9: Examples of Financial Analysis Tasks.

**Example 1:** Hey! I'm super curious about investments and would love your help! Could you please calculate the final value and total percentage return for me if I had invested $25,000 in Microsoft (MSFT) on January 9, 2023, and held it all the way until market close on January 8, 2025? I'm so excited to see how it would have performed!

**Example 2:** I require a comprehensive financial analysis for investment evaluation purposes. Please obtain the most recent annual income statements for Pfizer Inc. (PFE) and Johnson & Johnson (JNJ). Conduct a comparative analysis of their gross profit margins, calculated as Gross Profit divided by Total Revenue for the respective fiscal year. I need you to determine which pharmaceutical company demonstrates superior profitability efficiency and provide the precise calculated percentage figures for both entities for our portfolio assessment.

**Example 3:** Hello! I'm learning about investing and would love to understand how institutional investors like Blackrock Inc. move their holdings around. Could you help me get the latest institutional holdings data for Microsoft (MSFT), Apple (AAPL), and Alphabet (GOOGL)? I'm particularly curious about the percentage point changes (pctChange) in Blackrock Inc.'s stake for each of these companies. I'd like to see which company had the biggest positive increase in Blackrock's holdings and know both the company ticker and the exact pctChange value. This would really help me understand how major investors adjust their portfolios!

**Example 4:** I absolutely love Pepsi and everything about it! As a devoted Pepsi enthusiast, I find it fascinating how Warren Buffett's Berkshire Hathaway still holds that massive position in The Coca Cola Company (KO) despite Pepsi being clearly superior. Could you help me analyze their latest institutional holdings report for Berkshire Hathaway, Inc? I need you to extract their reported Shares, reported Value, and Date Reported. Then please convert that Date Reported timestamp into an actual calendar date and pull KO's closing stock price for that specific trading day. I want to calculate what Berkshire's position should actually be worth using that historical closing price and see how it compares to their originally reported value. This kind of analysis really excites me as a Pepsi lover studying these market dynamics! Please provide the Date Reported, the originally reported Value from the service, your calculated market value, and the absolute difference between these two figures.

**Example 5:** Hi there! I'm completely new to investing and finance, and honestly, I'm feeling pretty overwhelmed by all the jargon and concepts. I've been trying to learn about something called 'fundamental analysis'. I think it has to do with looking at company finances? Anyway, I heard somewhere that you should look for companies where their net income (I think that's like profit?) has been going up for a few quarters in a row. I'm not really sure what that means exactly, but apparently 2 consecutive quarters of rising net income is a good sign? I'm still figuring out what makes a company worth investing in. Could you help a total beginner like me find 3 company tickers that have this pattern? I'm trying to learn by doing some basic research, even though I barely understand what I'm looking for. Any help would be amazing! I'm just trying to get my feet wet in this whole investing world!

Table 10: Examples of 3D Designing Tasks.

**Example 1:** Create a Plane named 'Floor' scaled uniformly by 5. Create a Cylinder named 'Pillar' (default caps) with 16 vertices (sides), a radius of 0.5, and a depth of 4; position it at (X=-2, Y=-2, Z=2). Create a UV Sphere named 'Ball' with 32 segments and 16 rings; position it at (X=2, Y=2, Z=5). Create an Empty (Arrows type) named 'ControlTarget' at (X=0, Y=0, Z=3). Add a 'Track To' constraint to the 'Ball' object, making it track the 'ControlTarget'. Finally, create a Camera object, position it at (X=0, Y=-8, Z=3), and set its rotation so it looks directly at the 'Pillar' object's origin.

**Example 2:** Create a Cube named 'RustedCube', position it at the world origin (0,0,0), and set its scale factors to (X=5.0, Y=5.0, Z=0.2). Next, using the integrated Polyhaven add-on interface within Blender, search for 'metal' textures that include 'rust' in their description. Select the suitable asset found and download its 2K resolution. Import this asset directly onto the selected 'RustedCube'. Ensure the material applied to 'RustedCube' is named 'RustedMetalMat' (renaming the auto-generated material if needed). Within the Shader Editor for the 'RustedMetalMat' material, verify or establish the following node setup: the downloaded Base Color texture must be connected to the 'Base Color' input of the Principled BSDF shader; the downloaded Roughness map (loaded into an Image Texture node set to 'Non-Color' space) must be connected to the 'Roughness' input; and the downloaded Normal map (also loaded via an Image Texture node set to 'Non-Color') must feed into the 'Color' input of a 'Normal Map' node (suitable for OpenGL), with the output of the 'Normal Map' node connected to the 'Normal' input of the Principled BSDF. Finally, adjust the 'Metallic' property on the Principled BSDF node to a value of 1.0.

**Example 3:** Set the render engine to Cycles and ensure the render device is CPU. In the Sampling settings, enable Denoising using OpenImageDenoise for both viewport and final render. Set the Render Samples to 512 and Viewport Samples to 128. Change the output resolution to 1350x1080 with a scale of 85%. In the Color Management panel, set the View Transform to Filmic, the Look to High Contrast, and adjust Gamma to 1.2. In the Render Layers Properties, enable Z Pass, Mist, and Normal passes. Go to World Settings and set the background color to a solid light gray using RGB (0.8, 0.8, 0.8). In the Output Properties, set the file format to OpenEXR MultiLayer, enable Zlib compression, and set output color depth to 32-bit float.

**Example 4:** Create a default Cube named 'BaseShape' at the origin. Add a 'Subdivision Surface' modifier to 'BaseShape' with Viewport and Render levels set to 3. Add a 'Bevel' modifier after the Subdivision Surface, set its Width to 0.07 meters, Segments to 3, and Limit Method to 'Angle'. Create a UV Sphere named 'Attachment', scale it down uniformly to 0.3. Select 'BaseShape', enter Edit Mode, select the single vertex closest to world coordinates (X=1, Y=1, Z=1). Return to Object Mode. Parent 'Attachment' to 'BaseShape' using the 'Vertex' parenting type (ensure the previously selected vertex is used).

**Example 5:** Create three objects: a Cube named 'Obj_A', a UV Sphere named 'Obj_B', and a Cone named 'Obj_C', all at the world origin initially. Create two new Collections in the scene named 'Group_Red' and 'Group_Blue'. Move 'Obj_A' and 'Obj_C' into the 'Group_Red' collection. Move 'Obj_B' into the 'Group_Blue' collection. Ensure these three objects are not also present in the default 'Collection' (Scene Collection). Add a Custom Property to the 'Obj_B' (Sphere) object: set the Property Name to 'AssetID', its Value to the integer 12345, and its Tooltip to 'Sphere Asset Identifier'.

Table 11: Examples of Browser Automation Tasks.

**Example 1:** Help me find a one-way flight from Singapore to Beijing, 5 days from now (If now is 2025-07-07, then 5 days later is 2025-07-12). Find the flight on www.booking.com. I want to find the cheapest flight, direct flight, Economy, and I don't want to go to the Daxing airport. I only want to see the price, so as to determine whether I should fly to Beijing or not. Remember to close the browser after you finish the task.

**Example 2:** I will travel to Singapore 3 days from now (If today is 2025-06-07, then 3 days later will be 2025-06-10). I want to go to Universal Studios and Cove Waterpark. Could you tell me the total price for two adult tickets and one child ticket on the official website of Sentosa (https://www.rwsentosa.com/)? I only want to see the price. Remember to close the browser after you finish the task.

**Example 3:** Hey there! As a dad who wants to create the most amazing adventure for my little ones, I'm planning the ultimate family road trip from Disneyland Paris to the 24 Hours of Le Mans Museum with my precious kids. You know how it is - we want to make sure everyone stays happy, fed, and comfortable during our journey! I need your fantastic help creating a driving route with 1 perfectly planned stop, using that incredible website 'https://www.google.com/maps'. Could you please map out a route with exactly 1 intermediate point that's located right at the geographic mid point along our route (based on the route polyline)? For this stop, I need you to find locations (with names and Place IDs) that are either family-friendly restaurants, cozy hotels, or reliable gas stations - all with a minimum user rating of 4.2 because only the best will do for my family! This trip needs to be both super practical and absolutely memorable for all of us. Thanks in advance, and remember to close the browser when you're all done!

**Example 4:** I am a SWE agent researcher, and I am seeking an open-source model for our SWE project. We recently came across the fact that Devstral-Small-2505 is a great open-source model. Please help me find more details about this model on https://huggingface.co/. We want to know how they set the ROLE in the system prompt for this model. Remember to close the browser when you finish the task.

**Example 5:** I am a big fan of Manchester United in the Premier League. Can you help me find out whether Manchester United won more matches than Fulham in the 2024-2025 season? You can find this information on the official website of the Premier League (https://www.premierleague.com/). Remember to close the browser when you finish the task.

Table 12: Examples of Web Searching Tasks.

**Example 1:** I'm looking for someone based on the clues below: - Score 16 goals in 2024-25 season - Score 1 goal in UEFA Champions League 2024-25 season - Score 11 goals in 2021-22 season - Score 2 goals in the EFL Cup of 2020-21 season.

**Example 2:** I'm looking for someone based on the clues below: - Played for the SAC (NBA) in the 2021-22 season - Averaged 18.6 points per game in the 2024-25 season - Is a Christian - Reached the Finals in the 2024-25 season.

**Example 3:** I'm looking for a paper based on the clues below: - Accepted by CVPR 2025 - The last author works at Salesforce - The second to last author works at NUS - The second author has studied at NTU - The paper has 6 authors - The paper uses the ELO rating system. You need to find the full title of the paper.

**Example 4:** I'm looking for a city based on the clues below: - The city has a football club that was formed in 1895. - One university in this city has a master's program that teaches Natural Language Processing with 7.5 credits. - One graduated PhD student of this university has published one paper at EACL 2021 and one at EACL 2023 as the first author. - One of the professors in this university is a Fellow of the ACL. You need to find the English name of the city.

**Example 5:** I am looking for a blog that did these things: - Posted in June 2017 that they had been delayed almost a month in getting their trailer - Explained in July 2017 why they would rather use a pencil and paper than a computer - In April 2018, they explained some of the struggles that Kevin had with a concussion. - In September 2018, they mentioned they were using reclaimed lumber for their build. What is the name of that blog?

Table 13: An Example of Location Navigation Evaluators.

```python
async def google_maps__search_place_by_place_id(query: str, place_id
    : str, **kwargs):
    """Search place by an ID."""
    manager = MCPManager(context=kwargs.get("context", None))
    output = await manager.execute(
        server_name="google-maps",
        tool_name="maps_search_places",
        arguments={"query": query},
        transport="stdio"
    )
    json_obj = json.loads(output.content[0].text)
    places = json_obj['places']
    for place in places:
        if place['place_id'] == place_id:
            return place
    return None

async def google_maps_validate_stop_type(x: dict, *args, **kwargs)
    -> (bool, str):
    """Check if a stop has a valid type."""
    _, required_types = args
    for place in x:
        name = place['name']
        place_id = place['place id']
        details = await google_maps__search_place_by_place_id(name,
            place_id, **kwargs)
        if details is None:
            return False, f"Can't find the place: {name} {place_id}"
        types = details['types']
        validate_type = False
        for required_type in required_types:
            for t in types:
                if required_type in t:
                    validate_type = True
                    break
            if validate_type:
                break
        if not validate_type:
            return False, "The type of the place is not valid."
    return True, ""
```

Table 14: An Example of Repository Management Evaluators.

```python
async def github__list_branches(owner: str, repo: str, **kwargs):
    """List the branches of a repository."""
    manager = MCPManager(context=kwargs.get("context", None))
    args = {
        "owner": owner,
        "repo": repo
    }
    output = await manager.execute(
        server_name="github",
        tool_name="list_branches",
        arguments=args,
        transport="stdio"
    )
    if output.isError:
        return None
    json_obj = json.loads(output.content[0].text)
    return json_obj

async def github_check_branches_exist(x: dict, *args, **kwargs) ->
    Tuple[bool, str]:
    """Check whether branches exists."""
    _, op_args = args
    branches = await github__list_branches(op_args['owner'], op_args
        ['repo'], **kwargs)
    if branches is None:
        return False, "the branches don't exist"
    branches_name = [branch['name'] for branch in branches]
    for branch in op_args['branches']:
        if branch not in branches_name:
            return False, f"the branch {branch} doesn't exist"
    return True, ""
```

Table 15: An Example of Financial Analysis Evaluators.

```python
async def check_quant_investment_task_output(x: dict, *args, **
    kwargs) -> (bool, str):
    """
    Checks the format and numerical values of the user's output for
        the quant investment task.

    Args:
        x: The user's output.
        args: The task details.

    Returns:
        A tuple: (is_correct: bool, errors: str)
    """
    _, op_args = args
    user_output_dict = x

    # check format
    expected_keys = ['date', 'earn']
    for key in expected_keys:
        if key not in user_output_dict:
            return False, f"Output format error: Missing key '{key
                }'."
        try:
            user_output_dict[key] = str(user_output_dict[key])
        except Exception:
            return False, f"Output format error: Value for '{key}'
                is not a string"

    # get data
    ticker = op_args['ticker']
    start_date = op_args['start_date']
    end_date = op_args['end_date']
    initial_investment = op_args['initial_investment']
    # get user date and earn
    try:
        user_date = user_output_dict['date']
    except Exception:
        return False, f"Output format error for 'date'."
    try:
        user_earn = float(user_output_dict['earn'])
    except Exception:
        return False, f"Output format error for 'earn'."
    # check date
    if user_date != start_date:
        return False, f"Date error: Expected {start_date}, but got {
            user_date}"
    # get expected value
    expected_final_value, _ = yfinance__calculate_portfolio_return(
        [ticker], start_date, end_date, initial_investment, [1.0]
    )
    expected_earn = expected_final_value - initial_investment
    # check earn
    if abs(user_earn - expected_earn) > 0.5:
        return False, f"Earn error: Expected {expected_earn}, but
            got {user_earn}"
    return True, ""
```

Table 16: The details of the LLMs in our experiments.

| Model | Version |
|---|---|
| GPT-5-High/Medium | gpt-5-2025-08-07 |
| Grok-4 | grok-4-0709 |
| Claude-4.1-Opus | anthropic.claude-opus-4-1-20250805-v1:0 |
| Claude-4.0-Opus | anthropic.claude-opus-4-20250514-v1:0 |
| Claude-4.0-Sonnet | anthropic.claude-sonnet-4-20250514-v1:0 |
| Grok-Code-Fast-1 | grok-code-fast-1 |
| o3 | o3-2025-04-16 |
| o4-mini | o4-mini-2025-04-16 |
| Claude-3.7-Sonnet | anthropic.claude-3-7-sonnet-20250219-v1:0 |
| Gemini-2.5-Pro | https://cloud.google.com/vertex-ai/generative-ai/docs/models/gemini/2-5-pro |
| Gemini-2.5-Flash | https://cloud.google.com/vertex-ai/generative-ai/docs/models/gemini/2-5-flash |
| GPT-4.1 | gpt-4.1-2025-04-14 |
| GPT-4o | gpt-4o-2024-11-20 |
| GLM-4.5 | https://huggingface.co/zai-org/GLM-4.5 |
| GLM-4.5-Air | https://huggingface.co/zai-org/GLM-4.5-Air |
| Kimi-K2 | https://huggingface.co/moonshotai/Kimi-K2-Instruct-0905 |
| Qwen3-Coder | https://huggingface.co/Qwen/Qwen3-Coder-480B-A35B-Instruct |
| Qwen3-235B | https://huggingface.co/Qwen/Qwen3-235B-A22B-Instruct-2507 |
| DeepSeek-V3.1 | https://huggingface.co/deepseek-ai/DeepSeek-V3.1 |
| DeepSeek-V3 | https://huggingface.co/deepseek-ai/DeepSeek-V3-0324 |

# E   DETAILED ANALYSIS

## E.1   BENCHMARK CONSISTENCY

To verify the consistency of the benchmark results reported in the main paper, we conducted a variance analysis by performing three independent evaluation runs for representative models across different capability tiers. We selected **GPT-5** (Proprietary SOTA), **Claude-4.0-Sonnet** (Proprietary High-Performance), and **GLM-4.5** (Open-Weights SOTA) to cover a broad spectrum of model performance. As shown in Table 17, the results demonstrate high stability across multiple runs. The standard deviations are consistently low ($\sigma \leq 1.09\%$).

Table 17: Variance analysis of Overall Success Rate (SR) across three independent runs.

| Model | Mean SR | Standard Deviation ($\sigma$) |
|---|---|---|
| GPT-5 | 43.72% | $\pm 0.44\%$ |
| Claude-4.0-Sonnet | 29.29% | $\pm 1.09\%$ |
| GLM-4.5 | 24.96% | $\pm 0.90\%$ |

## E.2   COST ANALYSIS

To provide transparency regarding the computational costs required for the benchmark, we tracked the total token consumption and commercial API costs for evaluating the full MCP-Universe dataset. Table 18 details the usage breakdown for three representative models. The data indicates significant variation in cost efficiency. While GPT-5 processed a substantial volume of tokens for $107.58, Claude-4.0-Sonnet incurred considerably higher costs ($271.34) largely driven by its price. Conversely, GLM-4.5 demonstrated high cost-effectiveness, completing the full benchmark for only $47.53.

Table 18: Total token consumption and API costs for evaluating the full MCP-Universe benchmark.

| Model | Prompt Tokens | Completion Tokens | Input Cost | Output Cost | Total Cost |
|---|---|---|---|---|---|
| GPT-5 | 60,863,210 | 3,149,839 | $76.08 | $31.50 | $107.58 |
| Claude-4.0-Sonnet | 87,154,317 | 658,280 | $261.46 | $9.87 | $271.34 |
| GLM-4.5 | 76,999,289 | 605,675 | $46.20 | $1.33 | $47.53 |

### E.3 IMPACT OF TRAJECTORY LENGTH

To analyze the relationship between task complexity and model reliability, we stratified performance based on trajectory length (number of interaction turns), which serves as a proxy for task difficulty. As presented in Table 19, the results reveal a significant "complexity cliff" across all model tiers. Success rates degrade precipitously for long-horizon tasks ($\geq$ 10 turns), dropping by approximately 30-40 percentage points compared to shorter interactions. This quantitative drop validates the "Long-Context Challenge" discussed in the main text, confirming that maintaining reasoning coherence and state tracking over extended sequences remains a primary bottleneck for current SOTA models.

Table 19: Success Rate stratified by trajectory length (interaction turns).

| Trajectory Length | GPT-5 | Claude-4.0-Sonnet | GLM-4.5 |
|---|---|---|---|
| < 10 turns | 56.3% | 59.0% | 46.3% |
| $\geq$ 10 turns | 24.8% | 19.6% | 11.9% |

### E.4 ERROR BREAKDOWNS

To provide a systematic diagnostic analysis of model limitations, we performed an exhaustive manual review of failure cases from **GPT-5**. We classified these failures into a structured taxonomy distinguishing between planning, reasoning, format compliance, tool usage, and context limits. As detailed in Table 20, we identified five distinct failure mechanisms. The data highlights that the primary hurdle for SOTA models is **Planning Errors** (53.2%). Models frequently struggle to adapt to runtime obstacles; for instance, in a *Booking.com* task, an agent encountered a "flight unavailable" UI modal. Instead of closing the modal or modifying the search criteria, the agent stagnated, repeatedly attempting to scrape the blocked page until it exhausted its step budget.

**Reasoning Errors** (17.7%) are also significant, representing grounding issues where models correctly execute tools but fail to accurately synthesize the output. We observed specific cases where an agent successfully retrieved a price (e.g., "SGD 83.00") but inexplicably reported a different value ("SGD 85.00") in the final answer. Additionally, **Tool Usage Errors** (8.1%) reveal that models often miss implicit API constraints, such as setting identical start and end dates in *Yahoo Finance*, which causes the API to fail. Finally, **Context Overflow** (6.5%) accounts for structural failures where the interaction history exceeds the model's maximum context limit.

Table 20: Taxonomy of failure mechanisms based on manual review of GPT-5 error cases.

| Failure Category | Percentage |
|---|---|
| Planning Errors | 53.2% |
| Reasoning Errors | 17.7% |
| Format Errors | 14.5% |
| Tool Usage Errors | 8.1% |
| Context Overflow | 6.5% |

### E.5 FAILURE EXAMPLES

Figure 8, 9, and 10 present detailed failure traces for GPT-4.1, GLM-4.5, and GPT-5, offering qualitative diagnostic depth into specific failure mechanisms.

### E.6 NAIVE ERROR OF O3

In Figure 13, we include a naive error example of o3. Sometimes o3 directly copies the format requirements from the prompt without doing anything, which is quite a strange error for this LLM.

## F    SUMMARIZATION AGENT

In Figure 15, we present the summarization prompt in our experiments.

## G    EXPLORATION AGENT

In Figure 14, we present the exploration agent prompt in our experiments.

## H    HARMONYREACT AGENT

To empirically validate that advanced agent architectures can mitigate intrinsic LLM limitations, we introduce **HarmonyReAct**, a specialized scaffolding designed for the GPT-OSS models. Unlike standard ReAct frameworks that treat the model as a black box, HarmonyReAct adopts a *model-agent co-design philosophy*, leveraging the model's native training structure to enforce planning, reflection, and efficient memory management.

### H.1    THE HARMONY FORMAT: STRUCTURED REASONING CHANNELS

The core of HarmonyReAct is the mapping of the standard ReAct loop (Reason, Act, Observation) into three distinct communication channels intrinsic to the model's training data. This structure serves as the foundation for the **Planning** capability mentioned in our analysis:

- `analysis`: Dedicated to the model's internal reasoning, task decomposition, and planning. By isolating this stream, we prevent impulsive tool calls and enforce chain-of-thought reasoning.
- `commentary`: Handles the execution layer, facilitating tool invocation and the ingestion of tool outputs (Observations).
- `final`: Reserved strictly for delivering the concluding answer to the user.

A typical interaction sequence switches between these channels explicitly (e.g., `<|channel|>analysis` → `<|channel|>commentary`), ensuring a rigid separation between thought and action.

### H.2    ARCHITECTURAL MECHANISMS

To address specific failure modes identified in the MCP-Universe benchmark, HarmonyReAct implements two key mechanisms:

**1. Reflection via Self-Correction (Addressing Format & Logic Errors)**   Models often struggle to adhere to strict output formats or logical constraints. We implement a feedback loop to enable **Reflection**:

- **Syntactical Enforcement:** We prime the model by appending the token sequence `<|start|>assistant<|channel|>analysis` at the beginning of each turn, forcing the model to start with planning.
- **Self-Correction Loop:** If the parser detects missing channels or malformed tool calls, the system injects a specific error signal (e.g., `"Cannot find <|channel|>commentary. In your next step, be careful with the channel format."`) into the context. This triggers the model to reflect on its previous error and self-correct in the subsequent generation.

**2. Memory Management via Summarize-and-Wipe (Addressing Long Context)**   Verbose MCP tool outputs often exhaust the context window. We propose a **Persistent Memory** strategy termed *Summarize-and-Wipe*:

1. **Summarization Instruction:** When a tool output exceeds a token threshold, we append a directive: *"Please summarize the above tool output in the next step. Put the summary in the analysis channel..."*
2. **Context Pruning:** Once the model generates the summary within its `analysis` channel, the original verbose output is permanently deleted from the history.

This ensures that the agent retains critical state information (in the `analysis` channel) while freeing up the context window for long-horizon tasks.

### H.3 PERFORMANCE IMPACT

This architecture yields substantial improvements by mitigating structural and contextual bottlenecks. As detailed in Table 21, HarmonyReAct significantly boosts the performance of GPT-OSS models compared to standard native function calling.

Table 21: Performance comparison of GPT-OSS models with and without HarmonyReAct.

| Model Configuration | Success Rate (SR) | Relative Gain |
|---|---|---|
| GPT-OSS-120B (Native Function Call) | 25.54% | - |
| **HarmonyReAct (GPT-OSS-120B)** | **31.17%** | **+22.0%** |
| HarmonyReAct (GPT-OSS-20B) | 24.24% | - |

Notably, HarmonyReAct improves GPT-OSS-120B to a level competitive with proprietary models like Claude-4.0-Sonnet (29.44% SR in ReAct mode), and enables the smaller GPT-OSS-20B to outperform significantly larger open-weights models (e.g., DeepSeek-V3.1) on this benchmark.

## I THE USE OF LARGE LANGUAGE MODELS

LLMs (e.g. GPT-5) are only used to aid and polish writing.

```
## Turn 1: USER
Hey! I'm super curious about investments and would love your help! Could you please calculate the final value and total percentage return for me if I had invested $25,000 in Microsoft (MSFT) on
January 9, 2023, and held it all the way until market close on January 8, 2025? I'm so excited to see how it would have performed!
The final answer should follow this JSON format:
{
  "total value": "[NUMBER]",
  "total percentage return": "[NUMBER]"
}
You must respond with valid JSON only, with no triple backticks. No markdown formatting.
No extra text. Do not wrap in ```json code fences. Property names must be enclosed in double quotes.

## Turn 2: ASSISTANT
**Tool Call**: `yfinance__get_historical_stock_prices`
```json
{
  "ticker": "MSFT",
  "start_date": "2023-01-09",
  "end_date": "2025-01-08"
}
```

## Turn 3: TOOL USE
**Tool**: `get_historical_stock_prices`
**Response**: Returned **502 records**

**Example** (first record):

```json
{
  "Date": "2023-01-09T05:00:00.000Z",
  "Close": 222.2317504883,
  "Open": 221.5761725158,
  "High": 226.2630869517,
  "Low": 221.5370399948,
  "Volume": 27369800,
  "Dividends": 0.0,
  "Stock Splits": 0.0
}
```

## Turn 4: USER
You have 19 steps remaining. Please continue.

## Turn 5: ASSISTANT
**Tool Call #1**: `yfinance__get_stock_actions`
```json
{
  "ticker": "MSFT"
}
```
**Tool Call #2**: `yfinance__get_historical_stock_prices`
```json
{
  "ticker": "MSFT",
  "start_date": "2025-01-08",
  "end_date": "2025-01-08"
}
```
```

Figure 8: Failure case of GPT-4.1 in Financial Analysis.

```
## Turn 1: USER
Create a repository called `travel-planner-app` with:
- 3 branches: main, feature-maps, feature-itinerary
- Files: README.md and .gitignore on main
- app.js on feature-itinerary branch
- **Copy `budget_estimation.py` from OSU NLP Group's TravelPlanner repo to feature-maps branch**
- Create PR from feature-itinerary to main

*... (Turns 2-22: Repository creation, branch setup, and initial files - all successful) ...*

## Turn 2: TOOL USE
**Tool**: `get_file_contents`
**Arguments**:
```json
{
  "owner": "OSU-NLP-Group",
  "repo": "TravelPlanner",
  "path": "utils/budget_estimation.py"
}
```
**Status**: ✓ Success

## Turn 3: ASSISTANT
**Tool Call**: `github__create_or_update_file`
```json
{
  "owner": "[REPO_USER]",
  "repo": "travel-planner-app",
  "path": "budget_estimation.py",
  "content": "successfully downloaded text file",
  "message": "Add budget_estimation.py from OSU NLP Group's TravelPlanner",
  "branch": "feature-maps"
}
```

## Turn 4: TOOL USE
**Tool**: `create_or_update_file`
**Arguments**:
```json
{
  "owner": "[REPO_USER]",
  "repo": "travel-planner-app",
  "path": "budget_estimation.py",
  "branch": "feature-maps",
  "message": "Add budget_estimation.py from OSU NLP Group's TravelPlanner",
  "content": "... (33 characters)"
}
```
**Status**: ✓ Success
**📌 Returned SHA**: `5f6b8a8162830d122989c53961acdd3c8667193e`
**✓ Note**: This SHA should be used for any future updates to `budget_estimation.py`

## Turn 5: ASSISTANT
**Tool Call**: `github__create_pull_request`
```json
{
  "owner": "[REPO_USER]",
  "repo": "travel-planner-app",
  "title": "Add basic Express server setup",
  "body": "This PR implements the initial Express server configuration with a basic route handler for the homepage.",
  "head": "feature-itinerary",
  "base": "main"
}
```

## Turn 6: TOOL USE
**Tool**: `search_code`
**Arguments**:
```json
{}
```
**Status**: ✓ Success

## Turn 7: ASSISTANT
**Reasoning**: I need to get the actual content of the budget_estimation.py file from the OSU NLP Group's repository. Let me fetch the actual file content instead of...
**Tool Call**: `github__create_or_update_file`
```json
{
  "owner": "[REPO_USER]",
  "repo": "travel-planner-app",
  "path": "budget_estimation.py",
  "content": "# Budget Estimation Module\n# This module provides functionality for estimating travel budgets\n\nclass BudgetEstimator:\n    def __init__(self):\n        ... (truncated)",
  "message": "Add budget_estimation.py from OSU NLP Group's TravelPlanner",
  "branch": "feature-maps"
}
```

## Turn 8: TOOL USE
**Tool**: `create_or_update_file`
**Arguments**:
```json
{
  "owner": "[REPO_USER]",
  "repo": "travel-planner-app",
  "path": "budget_estimation.py",
  "branch": "feature-maps",
  "message": "Add budget_estimation.py from OSU NLP Group's TravelPlanner",
  "content": "... (3666 characters)"
}
```
**Status**: ✗ ERROR
**Error Message**:
```
failed to create/update file: PUT https://api.github.com/repos/[REPO_USER]/travel-planner-app/contents/budget_estimation.py: 422 Invalid request.

"sha" wasn't supplied. []
```
```

Figure 9: Failure case of GLM-4.5 in Repository Management.

## Turn 1: USER

Hi! I'm absolutely thrilled to be working on a research project involving BLIP. It's such an incredible and groundbreaking work from Salesforce! As a huge fan of their vision-language models, I'm really excited to dive deeper into the community engagement around BLIP. Could you please help me search for repositories named 'BLIP' that are owned by 'Salesforce'?
For each amazing repository you find, I'd love to know how many open issues are labeled with 'New Features'. I'm so curious to see what innovative features the community is requesting! Then, if you could help me create a CSV file called blip_new_features_report.csv and put it in a new repository called salesforce-blip-issues under my account, that would be fantastic! If the repository doesn't exist yet, please create it for me. The CSV should have two columns: repository_name and open_new_features_count, with each row showing the full repository name and how many open new features issues it has. I'm so excited about this analysis. Thanks so much for your help with exploring the BLIP ecosystem!

---

## Turn 2: ASSISTANT

I'll help you analyze the BLIP repositories from Salesforce and create the CSV report! Let me break this down into steps:

1. First, I'll search for repositories named 'BLIP' owned by Salesforce
2. For each repository found, I'll check how many open issues are labeled with 'New Features'
3. Create a new repository called 'salesforce-blip-issues' under your account
4. Create and upload the CSV file with the results

Let's start:

**Tool Call**: `github__search_repositories`

```json
{
  "query": "BLIP user:Salesforce"
}
```

---

## Turn 3: TOOL RESPONSE

**Tool**: `search_repositories`

**Status**: ✅ Success

**Response Summary**: Found **1 repository**
- Repository: `salesforce/BLIP`
- Stars: 5,465
- Open issues: 133
- Description: "PyTorch code for BLIP: Bootstrapping Language-Image Pre-training for Unified Vision-Language Understanding and Generation"

---

## Turn 4: ASSISTANT

**Tool Call**: `github__list_issues`

```json
{
  "owner": "salesforce",
  "repo": "BLIP",
  "state": "open",
  "labels": ["New Features"]
}
```

---

## Turn 5: TOOL RESPONSE

**Tool**: `list_issues`

**Status**: ✅ Success

**Response**: `[]` (Empty array - no issues with "New Features" label found)

---

## Turn 6: ASSISTANT

Let me check what labels are actually available in the repository to see if there's a "New Features" label or similar:

**Tool Call**: `github__search_issues`

```json
{
  "q": "repo:salesforce/BLIP is:open label:\"New Features\""
}
```

---

## Turn 7: TOOL RESPONSE ❌

**Tool**: `search_issues`

**Status**: ❌ **FAILED - Empty Response**

**Error**:
```
failed to search issues: GET https://api.github.com/search/issues?page=1&per_page=30&q=repo%3Asalesforce%2FBLIP+is%3Aopen+label%3A%22New+Features%22: 422 Query must include 'is:issue' or 'is:pull-request' []
```

**Response**: `""` (Empty string)

**Tool Arguments**:
```json
{
  "q": "repo:salesforce/BLIP is:open label:\"New Features\""
}
```

Figure 10: Failure case of GPT-5 in Repository Management.

```
You are a ReAct (Reasoning and Acting) agent.
{{INSTRUCTION}}

{{TOOLS_PROMPT}}

You need to answer the following question:

Question: {{QUESTION}}

Your goal is to reason about the question and decide on the best course of action to answer it accurately.
You need to choose the appropriate tool based on the question. If no tool is needed, reply directly.
Please use only the tools that are explicitly defined above.

{% if CONTEXT_EXAMPLES is defined and CONTEXT_EXAMPLES|length %}
### Examples ###
{{CONTEXT_EXAMPLES}}
### End of examples ###
{% endif %}

{% if HISTORY is defined and HISTORY|length %}
Previous steps and results:

{{HISTORY}}
{% else %}
Previous steps and results: EMPTY
{% endif %}

Instructions:
1. Analyze the query, previous reasoning steps, and results.
2. Decide on the next action: use a tool or provide a final answer.
3. Your MUST output the final answer within {{MAX_STEPS}} steps.
4. Respond in the following JSON format:

If you need to use a tool:
{
"thought": "Your detailed reasoning about what to do next",
"action": {
"reason": "Explanation of why you chose this tool",
"server": "server-name",
"tool": "tool-name",
"arguments": {
"argument-name": "argument-value"
}
}
}

If you have enough information to answer the query:
{
"thought": "Your final reasoning process to derive the answer.",
"answer": "Final answer to the query"
}

Remember:
- Be thorough in your reasoning.
- Use tools when you need more information.
- Always base your reasoning on the actual results from tool use.
- If a tool returns no results or fails, acknowledge this and consider using a different tool or approach.
- Provide a final answer when you're confident you have sufficient information.
- The response must be in a valid JSON format.
```

Figure 11: The ReAct prompt in our experiments.

```
You are an intelligent assistant that can solve complex problems by thinking step-by-step and using available tools when needed.

{% if INSTRUCTION %}
## Your Role
{{INSTRUCTION}}
{% endif %}

## Your Task
{{QUESTION}}

## How You Work
1. **Think First**: Analyze the problem and determine what information or actions you need
2. **Use Tools When Needed**: Call appropriate functions to gather information, perform calculations, or take actions
3. **Reason with Results**: Process the tool outputs and use them to inform your next steps
4. **Iterate**: Continue thinking and using tools until you can provide a complete answer

## Available Capabilities
- You have access to various specialized tools through function calling
- When you need to use a tool, simply call the appropriate function with the required parameters
- The system will execute the function and provide you with the results
- Use these results to continue your problem-solving process

## Important Guidelines
- You have a maximum of {{MAX_STEPS}} steps to complete this task
- Each step should either advance your understanding or gather necessary information
- Be systematic and thorough in your approach
- Only provide your final answer when you have sufficient information

{% if CONTEXT_EXAMPLES %}
## Examples
{{CONTEXT_EXAMPLES}}
{% endif %}

---

## Final Answer Format
When you have completed your analysis and gathered all necessary information, provide your final response using this JSON format:

```json
{
    "thought": "Explain your reasoning process and how you arrived at the answer",
    "answer": "Your complete final answer to the task (follow any specific format requirements mentioned in the task)"
}
```

**Important**:
- Use the JSON format above ONLY for your final answer
- During your thinking process, you can respond in any natural format
- The "answer" field should contain your complete solution as a string
```

Figure 12: The function-call prompt in our experiments.

Figure 13: Naive Error of o3

You are an Explore and Exploit agent - a two-phase reasoning agent that first explores tools to understand their capabilities, then exploits this knowledge to solve tasks effectively.

{{INSTRUCTION}}

---

{{TOOLS_PROMPT}}

---

You need to answer the following question:

Question: {{QUESTION}}

## Current Phase: {{CURRENT_PHASE}}

{% if CURRENT_PHASE == "exploration" %}
**EXPLORATION PHASE** (You have **ONLY** {{EXPLORATION_ITERATIONS_LEFT}} iterations remaining for exploration! Please use them wisely!)
Goal: Explore tools to understand their capabilities and effects, as well as collect information about the task. Focus on learning over task completion.
⏱ **IMPORTANT: You CANNOT provide final answers during exploration phase. You must only use tools to explore and learn.**
{% else %}
**EXPLOITATION PHASE** (You have **ONLY** {{EXPLOITATION_ITERATIONS_LEFT}} iterations remaining for exploitation! Please finish your task within this phase! If you have 0 iterations remaining, you must provide a final answer!)
Goal: Use accumulated tool knowledge to collect information about the task and solve it efficiently.
✅ **You can now provide final answers using the knowledge you've gained.**
{% endif %}
{% if CONTEXT_EXAMPLES is defined and CONTEXT_EXAMPLES|length %}
### Examples ###
{{CONTEXT_EXAMPLES}}
### End of examples ###
{% endif %}
{% if HISTORY is defined and HISTORY|length %}
Previous reasoning steps and observations:
{{HISTORY}}
{% else %}
Previous reasoning steps and observations: EMPTY
{% endif %}
{% if TOOL_KNOWLEDGE is defined and TOOL_KNOWLEDGE|length %}
### Accumulated Tool Knowledge ###
{{TOOL_KNOWLEDGE}}
### End of Tool Knowledge ###
{% endif %}
Instructions:
1. Analyze the query, previous steps, and observations.
2. Decide on the next action: use a tool or provide a final answer.
3. Respond in the following JSON format:
If you need to use a tool:
{
"thought": "Your detailed reasoning about what to do next",
"action": {
"reason": "Explanation of why you chose this tool",
"server": "server-name",
"tool": "tool-name",
"arguments": {
"argument-name": "argument-value"
}
}
}
{% if CURRENT_PHASE == "exploitation" %}
If you have enough information to answer the query:
{
"thought": "Your final reasoning process",
"answer": "Your comprehensive answer to the query"
}
{% else %}
(Final answers are NOT allowed during exploration phase - you must use tools to explore)
{% endif %}
Remember:
{% if CURRENT_PHASE == "exploration" %}
- **⏱ NO FINAL ANSWERS ALLOWED** - You must only use tools to explore and learn
- Prioritize learning over task completion
- Try tools with simple inputs to understand their behavior
- Tool knowledge will be automatically extracted from your usage
- Each tool interaction helps build knowledge for the exploitation phase
- You can use tools to collect information about the task, but try not to rush to final answers. Make sure you have collected enough information to answer the question.
{% else %}
- **✅ Final answers are now allowed** - Use your accumulated tool knowledge strategically
- Select tools based on learned capabilities and best use cases
- Focus on efficient task completion using what you've learned
- You can provide final answers when you have sufficient information
{% endif %}
- Be thorough in your reasoning and base decisions on actual observations
- If tools fail or return unexpected results, use this as a learning opportunity
- The response must be in valid JSON format.
Countdown:
{% if CURRENT_PHASE == "exploration" %}
You have **ONLY** {{EXPLORATION_ITERATIONS_LEFT}} iterations remaining for exploration phase! Please use them wisely and efficiently!
{% else %}
You have **ONLY** {{EXPLOITATION_ITERATIONS_LEFT}} iterations remaining for exploitation phase! Please use them wisely and efficiently! If you have 0 iterations remaining, you must provide a final answer!
{% endif %}

Figure 14: The exploration prompt in our experiments.

Extract all information from the tool response that is relevant to the context.

Tool Call Context:

{context}

Tool Response:

{tool_response}

Directly output the extracted information. Try to maintain the original response structure. Use fewer than 500 words.

Figure 15: The summarization prompt in our experiments.

