# OpenReview forum: "MCP-Universe: Benchmarking Large Language Models with Real-World Model Context Protocol Servers"
_ICLR.cc/2026/Conference — Submitted to ICLR 2026_

### Official Review · Reviewer_eTRd · 2025-10-16

**Soundness:** 3
**Presentation:** 3
**Contribution:** 3
**Rating:** 6
**Confidence:** 5

**Summary:**

This paper introduces MCP-Universe, a benchmark designed to evaluate Large Language Models (LLMs) on complex, realistic tasks using real-world Model Context Protocol (MCP) servers. The authors created this benchmark to address the limitations of existing evaluations, which they argue are too simplistic and fail to capture real-world challenges such as long-horizon reasoning and interaction with large, unfamiliar toolsets.

The benchmark consists of 231 tasks across 6 domains: Location Navigation, Repository Management, Financial Analysis, 3D Design, Browser Automation, and Web Searching.
It utilizes 11 live MCP servers, including Google Maps, GitHub, and Yahoo Finance, which provide 133 distinct tools for the models to use.
To ensure rigorous and objective results, the framework uses execution-based evaluators instead of relying on an "LLM-as-a-judge" approach. These are divided into format, static, and dynamic evaluators that can check for compliance, time-invariant content, and real-time ground truth.
The evaluation framework is open-source, allowing researchers to seamlessly integrate new LLMs, agents, and MCP servers.

Even top-performing models like GPT-5-High and Grok-4 achieved low success rates of only 44.16% and 33.33%, respectively. This indicates a major gap between the general capabilities of LLMs and their effectiveness in real-world MCP environments.
Specialized, enterprise-level agents like Cursor and Claude Code did not achieve better performance than the standard ReAct framework.
A model's performance can be highly sensitive to its interaction paradigm. For example, the success rate of GPT-OSS-120B increased dramatically from 11.26% to 25.54% when switching from the OpenAI Agent SDK to a direct function-calling approach.

**Strengths:**

The paper's primary strength lies in its originality. It introduces MCP-Universe, which it posits as the first comprehensive benchmark for evaluating LLMs in real-world Model Context Protocol (MCP) environments. While other MCP-related benchmarks exist, this work distinguishes itself by moving beyond simplistic or derivative tasks (like adapting existing math or code datasets) and focusing on live, authentic MCP servers.

The research is of high quality, demonstrated by its methodological rigor. The benchmark's design is comprehensive, covering 6 diverse domains and 11 real-world MCP servers. A key strength is the development of a robust, execution-based evaluation framework that avoids the known biases of "LLM-as-a-judge" methods. By implementing distinct format, static, and dynamic evaluators, the authors ensure objective and reproducible assessments, which is especially critical for tasks involving time-sensitive data. The extensive experiments, testing over 20 leading LLMs with different agentic frameworks, further underscore the work's thoroughness.

The paper is exceptionally clear and well-structured. The authors effectively motivate the work by articulating the precise limitations of prior benchmarks. The architecture of the MCP-Universe framework is explained logically and supported by clear diagrams that illustrate the workflow from task configuration to evaluation. Results are presented in accessible tables and figures that cleanly summarize key performance metrics and comparisons across models and domains. The central arguments and contributions are stated explicitly, making the paper easy to follow and its conclusions straightforward to understand.

The work is highly significant as it addresses an urgent need within the AI community. With MCP rapidly becoming a standard for integrating LLMs with external tools, a reliable benchmark is crucial for measuring progress. The paper's findings are impactful, revealing that even state-of-the-art models like GPT-5-High (44.16% success rate) and Grok-4 (33.33% success rate) struggle significantly, highlighting that real-world tool use remains a major challenge. By identifying critical failure points such as long-context handling and unfamiliar tool usage, the research provides clear and actionable directions for future work in agent architecture and model development. The open-sourced, extensible framework is a valuable resource that can catalyze further research and standardization in the field.

**Weaknesses:**

The tasks were "manually designed" by the authors. While curated for difficulty, this approach risks a lack of diversity in the underlying problem structures within each domain. For instance, several Location Navigation tasks focus on finding optimal stopping points or equidistant locations. This may not fully represent the breadth of real-world geographic reasoning tasks. A more systematic approach, perhaps incorporating tasks derived from real user queries or logs from applications that use these tools, could ensure greater diversity and real-world representativeness.

The methodology explicitly discards tasks that can be "consistently completed within five retries". While this ensures the benchmark is challenging, it may also filter out the more common, simpler tasks that form the bulk of real-world agent interactions. This selection bias could skew the benchmark's results, making it an assessment of performance on "hard cases" rather than a representative measure of general utility. Including a separate track for more common, moderate-difficulty tasks would provide a more balanced perspective.

The primary metric is a binary success rate (SR), which treats a near-perfect solution that fails on a minor detail the same as a complete failure. The paper includes an "average evaluator score" (AE), but the relationship between partial success (high AE) and outright failure (low SR) is not deeply analyzed. For example, the paper notes that Claude-4.0-Sonnet achieves a higher AE than Grok-4 but a lower SR. A deeper analysis of why this occurs—such as identifying specific "killer" evaluators that cause catastrophic failure—would be highly instructive.

The evaluation focuses on correctness and the number of steps but overlooks crucial efficiency metrics like API costs, total token usage, or latency. In practical applications, an agent's cost-effectiveness and speed are as important as its ability to complete a task. An agent that succeeds but requires an exorbitant number of tokens or expensive tool calls may be impractical. Incorporating efficiency metrics would make the benchmark more aligned with real-world deployment constraints.

The paper identifies broad challenges like "long context" and "unknown tools". However, it would be more impactful to provide a detailed, quantitative breakdown of error types. For instance, when a model fails, is it due to incorrect tool selection, faulty parameter generation, misunderstanding the tool's output, or flawed multi-step planning? While a specific error example is given for Yahoo Finance, a systematic error analysis across all domains would provide more actionable insights for improving model reasoning and agent design.

The experiments primarily compare the ReAct framework with a native function-calling mode. While these are foundational, the field of autonomous agents includes other prominent paradigms like planning-and-solving or reflection-based methods. Testing a broader range of agent architectures on the benchmark would provide a more complete picture of which reasoning structures are best suited for complex, real-world tool use and why.

**Questions:**

Questions:

Could you elaborate on the methodology used to ensure task diversity and real-world representativeness, beyond simply curating for difficulty? Was there a systematic framework guiding task creation to avoid potential homogeneity?
How does filtering out easier tasks (solvable within five retries) affect the benchmark's generalizability? Does this focus on "hard cases" risk misrepresenting a model's typical real-world utility?
You note that some models achieve high partial scores (AE) but low final success rates (SR). Could you provide a deeper analysis of this discrepancy? Are specific evaluation steps consistently causing catastrophic failures?
Could you provide a quantitative, systematic breakdown of the most common error types (e.g., incorrect tool selection, faulty parameter generation, planning failures) for the top-performing models across all domains?
Were efficiency metrics such as API costs, token usage, or latency tracked during experiments? If so, could you discuss their implications, as these are critical for real-world deployment?

Suggestions:
Consider reporting on the distribution of partial scores (AE) for failed tasks. This could offer a more nuanced view of model capabilities by showing which models get "closer" to a correct solution.
The paper would be strengthened by including experiments with a wider range of agentic frameworks beyond ReAct and function calling, such as planning- or reflection-based agents, to provide a more comprehensive analysis of which reasoning structures excel.
To maximize the benchmark's impact, please consider releasing detailed setup instructions and containerized environments (e.g., Docker files) for the custom MCP servers, which would significantly lower the barrier for other researchers to reproduce and build upon your work.

---

> ### Author Response · Authors · 2025-11-26
> **Response to Reviewer eTRd [1/2]**
>
> Thank you for your thoughtful reviews! Your feedback provides valuable guidance for strengthening our paper. We are especially pleased that you view our work as highly significant and addressing an urgent need within the AI community.
>
> > **Weakness 1**: The tasks were "manually designed" by the authors. While curated for difficulty, this approach risks a lack of diversity in the underlying problem structures within each domain. For instance, several Location Navigation tasks focus on finding optimal stopping points or equidistant locations. This may not fully represent the breadth of real-world geographic reasoning tasks. A more systematic approach, perhaps incorporating tasks derived from real user queries or logs from applications that use these tools, could ensure greater diversity and real-world representativeness.
>
> **First**, regarding task diversity, Figure 3 demonstrates our broad coverage across 6 primary domains, further granularized into 27 sub-domains. Notably, the distribution is highly balanced, with no single sub-domain exceeding 9% of the total tasks. Addressing the specific example raised, tasks focusing on "optimal stopping points" comprise only 8.9% of the dataset, confirming that the benchmark avoids over-indexing on any single problem structure.
>
> **Second**, regarding real-world representativeness, all tasks are grounded in authentic user intents. The authors acted as real users to generate natural queries, and every task underwent rigorous manual verification to ensure it reflects the actual complexity of real-world tool usage.
>
> > **Weakness 2**: The methodology explicitly discards tasks that can be "consistently completed within five retries". While this ensures the benchmark is challenging, it may also filter out the more common, simpler tasks that form the bulk of real-world agent interactions. This selection bias could skew the benchmark's results, making it an assessment of performance on "hard cases" rather than a representative measure of general utility. Including a separate track for more common, moderate-difficulty tasks would provide a more balanced perspective.
>
> **First**, we emphasize that all selected tasks reflect **common real-world use cases**, such as making travel plans, checking ticket prices, and obtaining stock data. These are standard user intents rather than obscure "corner cases".
>
> **Second**, regarding the exclusion of trivial tasks, we argue that the core purpose of a benchmark is to **identify where current models fail**. If a benchmark consists of tasks that are easily solved by existing models, it loses its utility for distinguishing capabilities and driving progress. Therefore, we intentionally calibrate the difficulty to focus on the frontier of current agentic limitations.
>
> > **Weakness 3**: The primary metric is a binary success rate (SR), which treats a near-perfect solution that fails on a minor detail the same as a complete failure. The paper includes an "average evaluator score" (AE), but the relationship between partial success (high AE) and outright failure (low SR) is not deeply analyzed. For example, the paper notes that Claude-4.0-Sonnet achieves a higher AE than Grok-4 but a lower SR. A deeper analysis of why this occurs—such as identifying specific "killer" evaluators that cause catastrophic failure—would be highly instructive.
>
> We attribute the discrepancy between Higher AE and Lower SR to the **"Last-Mile Failure"** phenomenon, where models successfully execute intermediate steps (accumulating high AE) but fail strict, binary "Killer Evaluators" at the final stage. A prime example occurs in Location Navigation, where the elevation calculation requires precise alignment with specific address coordinates. Our analysis reveals that Claude-4.0-Sonnet often successfully plans and navigates to the target area (scoring high on process evaluators) but fails to align its final answer strictly with the Ground Truth required by the elevation calculator, leading to a 0% Success Rate. This divergence highlights that while AE captures **process robustness**, SR is essential for measuring **strict outcome reliability**, justifying the inclusion of both metrics.

---

> ### Author Response · Authors · 2025-11-26
> **Response to Reviewer eTRd [2/2]**
>
> > **Weakness 4**: The evaluation focuses on correctness and the number of steps but overlooks crucial efficiency metrics like API costs, total token usage, or latency. In practical applications, an agent's cost-effectiveness and speed are as important as its ability to complete a task. An agent that succeeds but requires an exorbitant number of tokens or expensive tool calls may be impractical. Incorporating efficiency metrics would make the benchmark more aligned with real-world deployment constraints.
>
> To address this, **we have added a comprehensive Cost and Token Analysis in Appendix E.2**. The table below reports the total token consumption and commercial API costs for evaluating the full benchmark across three representative models (representing both SOTA proprietary or open-source LLMs).
>
> | Model  | Prompt Tokens | Completion Tokens | Input Cost | Output Cost | Total Cost |
> | ----- | ----- |----- |----- |----- |----- |
> | GPT-5 | 60,863,210 | 3,149,839 | $76.08 | $31.50 | $107.58 |
> | Claude-4.0-Sonnet | 87,154,317 | 658,280 | $261.46 | $9.87 | $271.34 |
> | GLM-4.5 | 76,999,289 | 605,675 | $46.20 | $1.33 | $47.53 |
>
> > **Weakness 5**: The paper identifies broad challenges like "long context" and "unknown tools". However, it would be more impactful to provide a detailed, quantitative breakdown of error types. For instance, when a model fails, is it due to incorrect tool selection, faulty parameter generation, misunderstanding the tool's output, or flawed multi-step planning? While a specific error example is given for Yahoo Finance, a systematic error analysis across all domains would provide more actionable insights for improving model reasoning and agent design.
>
> To address this, we performed an **exhaustive manual review of failure cases** from GPT-5, categorizing them into a structured taxonomy that directly quantifies the error types you requested (e.g., planning vs. parameter generation).
>
> | Failure Category | Percentage |
> |-----------------|------------|
> | Planning Errors | 53.2% |
> | Reasoning Errors | 17.7% |
> | Format Errors | 14.5% |
> | Tool Usage Errors | 8.1% |
> | Context Overflow | 6.5% |
>
> This systematic analysis yields critical actionable insights: the dominant bottleneck is Planning (53.2%), specifically a lack of adaptive resilience. For instance, in a Booking.com task, the agent stagnated when facing a "flight unavailable" instead of modifying its search, revealing a specific deficit in multi-step error recovery. Furthermore, Reasoning failures (17.7%) highlight a disconnect in grounding, where models correctly execute tools but hallucinate the final answer (e.g., retrieving "SGD 83.00" but reporting "SGD 85.00"), while Tool Usage errors often stem from missing implicit API constraints, such as setting identical start and end dates in Yahoo Finance. This breakdown (**fully detailed in Appendix E.4**) clarifies that future agent design must prioritize planning, reasoning, format adherence, tool usage, and context management capabilities.
>
> > **Weakness 6**: The experiments primarily compare the ReAct framework with a native function-calling mode. While these are foundational, the field of autonomous agents includes other prominent paradigms like planning-and-solving or reflection-based methods. Testing a broader range of agent architectures on the benchmark would provide a more complete picture of which reasoning structures are best suited for complex, real-world tool use and why.
>
> We have extended our evaluation beyond foundational frameworks to include **enterprise-level agents** (Cursor Agent, Claude Code) in Figure 4 and introduced HarmonyReAct, an **improved agent detailed in the Appendix H**. HarmonyReAct specifically integrates the planning and reflection paradigms you requested by employing a model-agent co-design philosophy: it utilizes distinct reasoning channels to enforce structured planning and incorporates a self-correction mechanism that allows the model to reflect on and fix format errors. This advanced architecture improved GPT-OSS-120B's performance from 25.54% to 31.17%, empirically demonstrating that architectures incorporating explicit planning and reflection significantly outperform standard baselines on MCP-Universe.
>
> > **Questions**
>
> Please refer to our Response to Weakness above, where we have addressed all questions in detail.

---

### Official Review · Reviewer_Wvi6 · 2025-10-19

**Soundness:** 3
**Presentation:** 3
**Contribution:** 2
**Rating:** 4
**Confidence:** 4

**Summary:**

This work introduces a realistic benchmark for evaluating LLM agents with real MCP servers. The benchmark covers 6 everyday domains and 231 real-life tasks. Evaluation is human-written, rule-based, and execution-driven (format/static/dynamic), avoiding LLM-as-judge hallucination. Extensive experiments on current LMs show limited MCP-use and task completion capability. The authors further analyze long-context effects and irrelevant/unknown-tool issues, with ablations (e.g., summarization) revealing mixed improvements across domains.

**Strengths:**

1. The benchmark is timely and well designed: it uses real MCP servers and real-world, multi-turn agentic tasks rather than simulators.
2. The rule-based evaluation is solid and labor-intensive, avoiding typical LLM-as-judge pitfalls.
3. The evaluation dimensions are clear—domains and format/static/dynamic—and the study evaluates a broad set of models, yielding clear, differentiated results.
4. The exploratory analyses are useful, including long-context growth, unknown-tool misuse, and the impact of irrelevant servers. Letting the model do a few turns of exploration is a good perspective.
5. The writting is clear.

**Weaknesses:**

1. Insufficient analysis.
a) Lack of systematic error analysis. The paper would benefit from a structured taxonomy of failure causes across models and domains, with representative error cases. Format errors are only one category; others such as include tool selection, parameter filling, state tracking/memory failures. Such analysis would clarify the specific capability deficits and limitations of current models.
b) Unknown-tools exploration lacks case studies (Sec. 4.3). A focused set of interaction traces would help distinguish “tool unfamiliarity” from “API/argument misuse” or poor timing, and would ground the quantitative trends.

2. Metric coverage is somewhat limited. The primary emphasis on success rate omits other useful axes. Consider reporting cost (tokens/API calls), test time, performance stratified by task difficulty/trajectory length, and known vs. completely unseen tools. These would make the results more actionable for both research and deployment.

3. Prompt/MCP definition sensitivity. Different models may favor different prompt formats and tool-call schemas (e.g., GPT vs. Claude). It is unclear whether the authors tried alternative prompt templates or MCP definition variants to control for such biases. A brief sensitivity study—or at least documenting the chosen formats and any model-specific tuning—would strengthen fairness and interpretability.

**Questions:**

1. Questions in the weakness section.
2. Please report general testing time and cost, at least for one or two models.
3. Sec. 4.3 Long-context challenges: Are there failures purely due to over-long context? In Finance you argue fine-grained details are critical, yet GPT shows little change with/without summarization—how should we interpret this contrast with the big gains in Location Navigation?
4. Sec. 4.3 Unknown tools challenges: Why isn’t added exploration consistently helpful? In theory a few exploratory turns can be view as few shots for the tool and it should help model understand.
5. Which MCP servers are public/known vs. previously unseen to the models? This distinction could clarify tool-use generalization.

---

> ### Author Response · Authors · 2025-11-26
> **Response to Reviewer Wvi6 [1/2]**
>
> Thank you for your thoughtful reviews! Your feedback offers important direction for strengthening our paper. We are especially pleased that you consider our work both timely and well designed.
>
> > **Weakness 1**: Insufficient analysis. a) Lack of systematic error analysis. The paper would benefit from a structured taxonomy of failure causes across models and domains, with representative error cases. Format errors are only one category; others such as include tool selection, parameter filling, state tracking/memory failures. Such analysis would clarify the specific capability deficits and limitations of current models.
>
> We performed an **exhaustive manual review** of failure cases from GPT-5 to **provide the requested systematic error taxonomy**. As detailed below, we identified five distinct failure types:
>
> | Failure Category | Percentage |
> |-----------------|------------|
> | Planning Errors | 53.2% |
> | Reasoning Errors | 17.7% |
> | Format Errors | 14.5% |
> | Tool Usage Errors | 8.1% |
> | Context Overflow | 6.5% |
>
> The data highlights that the primary hurdle is Planning Error (53.2%). Models struggle to adapt to runtime obstacles. For instance, in a Booking.com task, the agent encountered a 'flight unavailable' pop-up. Instead of closing the page or modifying the search, it stagnated, repeatedly attempting to scrape the blocked page until it exhausted its step budget. Reasoning failures are also significant; we observed cases where the agent successfully retrieved a price ('SGD 83.00') but inexplicably reported a different number ('SGD 85.00') in the final answer. Additionally, Tool Usage errors reveal that models often miss implicit API constraints, such as setting identical start and end dates in Yahoo Finance, which causes the API to fail. Finally, Context Overflow accounts for failures where the interaction history exceeds the model's maximum context limit. **We have added the error breakdowns in the Appendix E.4 of the paper.**
>
> > **Weakness 1**: b) Unknown-tools exploration lacks case studies (Sec. 4.3). A focused set of interaction traces would help distinguish “tool unfamiliarity” from “API/argument misuse” or poor timing, and would ground the quantitative trends
>
> We have added specific interaction traces in **Appendix E.5**. These cases confirm that "tool unfamiliarity" manifests primarily as API/argument misuse (e.g., invalid date ranges in Yahoo Finance).
>
> > **Weakness 2**: Metric coverage is somewhat limited. The primary emphasis on success rate omits other useful axes. Consider reporting cost (tokens/API calls), test time, performance stratified by task difficulty/trajectory length, and known vs. completely unseen tools. These would make the results more actionable for both research and deployment.
>
> > **Question 2**: Please report general testing time and cost, at least for one or two models.
>
> We provide the requested additional metrics below, including a detailed cost breakdown and performance stratification by trajectory length.
>
> 1. **Cost Analysis**
>
> Table below reports the total token consumption and API costs for evaluating the full benchmark. **We have added this table in the Appendix E.2 of the paper.**
>
> | Model  | Prompt Tokens | Completion Tokens | Input Cost | Output Cost | Total Cost |
> | -----  | ----- | ----- | -----  | ----- | ----- |
> | GPT-5 | 60,863,210 | 3,149,839 | $76.08 | $31.50 | $107.58 |
> | Claude-4.0-Sonnet | 87,154,317 | 658,280 | $261.46 | $9.87 | $271.34 |
> | GLM-4.5 | 76,999,289 | 605,675 | $46.20 | $1.33 | $47.53 |
>
>
> 2. **Performance Stratified by Trajectory Length**
>
> We use trajectory length as a proxy for task difficulty. The results show a significant performance degradation for long-horizon tasks across all models. **We have added this table in the Appendix E.3 of the paper.**
>
> | Trajectory Length | GPT-5 | Claude-4.0-Sonnet | GLM4.5 |
> |-------------------|-------|-------|-------|
> | < 10 turns | 56.3% | 59.0% | 46.3% |
> | >= 10 turns| 24.8% | 19.6% | 11.9% |
>
> 3. **Regarding Known vs. Unseen Tools**
> A quantitative comparison is impossible due to **the closed-source nature of all models' training data**. While our analysis of specific failures (e.g., parameter misuse in GitHub Search and Yahoo Finance) strongly suggests that models are unfamiliar with certain MCP usage, we cannot definitively separate tasks into 'known' vs. 'unseen' categories without access to the models' training corpora.

---

> ### Author Response · Authors · 2025-11-26
> **Response to Reviewer Wvi6 [2/2]**
>
> > **Weakness 3**: Prompt/MCP definition sensitivity. Different models may favor different prompt formats and tool-call schemas (e.g., GPT vs. Claude). It is unclear whether the authors tried alternative prompt templates or MCP definition variants to control for such biases. A brief sensitivity study—or at least documenting the chosen formats and any model-specific tuning—would strengthen fairness and interpretability.
>
> To ensure fairness and control for bias, we strictly aligned all prompt formats and tool-call schemas with the official recommendations for each specific model. We explicitly assessed sensitivity by evaluating **two distinct strategies**: ReAct, which embeds tool definitions in the prompt, and Function Calling, which utilizes the model's native interface. Table 3 and Figure 4 quantify the performance differences between these approaches, and we have documented the exact model-specific templates in the Appendix C.
>
> > **Question 3**: Sec. 4.3 Long-context challenges: Are there failures purely due to over-long context? In Finance you argue fine-grained details are critical, yet GPT shows little change with/without summarization—how should we interpret this contrast with the big gains in Location Navigation?
>
> We clarify that the "Long Context" challenge encompasses two distinct failure modes: **hard Context Overflow** (accounting for 6.5% of total errors) and **performance degradation within valid limits**. In the latter case, Planning (53.2%) and Reasoning (17.7%) failures are influenced by the extended context. Regarding the contrast in summarization effectiveness, we attribute this to **information density**: Financial contexts are densely packed with critical numerical data, where lossy compression via summarization inevitably leads to precision loss and downstream errors. Conversely, Location Navigation contexts feature lower information density and structured cues, allowing the summarization agent to effectively retain key waypoints without losing decision-critical signals, which explains the performance gains in that domain.
>
> > **Question 4**: Sec. 4.3 Unknown tools challenges: Why isn’t added exploration consistently helpful? In theory a few exploratory turns can be view as few shots for the tool and it should help model understand.
>
> We clarify that exploration is indeed effective, evidenced by significant gains in Financial Analysis (Claude-4.0-Sonnet w/ +7.50%) and Browser Automation (GPT-4.1 w/ +7.69%). However, the lack of universal consistency stems from the inherent trade-off between information gain and context noise: while exploration generates valuable "few-shot" interaction traces, it simultaneously burdens the context window with intermediate steps that can distract the model from the primary instruction. Furthermore, given that our failure analysis identifies Planning (GPT-5, 53.2%), not Tool Usage, as the dominant failure mode, the marginal benefit of improved tool familiarity is often outweighed by the cognitive overhead of processing this extended, noisier context, explaining why gains are domain-dependent rather than universal.
>
> > **Question 5**: Which MCP servers are public/known vs. previously unseen to the models? This distinction could clarify tool-use generalization.
>
> A quantitative distinction between "known" and "unseen" MCP servers is infeasible due to the nature of proprietary training data; however, our empirical failure analysis strongly implies that models treat some MCP interfaces as unseen schemas. Notably, even SOTA models like GPT-4.1 and Claude-4.0-Sonnet frequently commit fundamental schema violations on tools for well-known platforms, such as violating date constraints in Yahoo Finance or misusing parameters in GitHub (e.g., passing invalid arguments to search_repositories or failing to initialize a README in create_repository). These systematic failures demonstrate that while models possess conceptual knowledge of the services, they lack specific priors for the exact MCP definitions used.

---

> > ### Comment · Reviewer_Wvi6 · 2025-11-26
> >
> > Thanks for the detailed response. My primary issues are resolved; I have raised my score.

---

> > > ### Author Response · Authors · 2025-11-26
> > > **Thank You**
> > >
> > > Thank you for reviewing our paper and raising the rating. We are glad your primary issues are resolved. We genuinely appreciate the time and thoughtful consideration you have dedicated to our work.

---

### Official Review · Reviewer_ooGm · 2025-10-27

**Soundness:** 3
**Presentation:** 3
**Contribution:** 2
**Rating:** 6
**Confidence:** 3

**Summary:**

The paper “MCP-Universe: Benchmarking Large Language Models with Real-World Model Context Protocol Servers” introduces a comprehensive benchmark to evaluate large language models (LLMs) in realistic, tool-connected environments via the Model Context Protocol (MCP). Unlike prior works that rely on simulated tool-use or GUI interactions, MCP-Universe connects directly to 11 real-world MCP servers across six domains — including Location Navigation, Repository Management, Financial Analysis, 3D Design, Browser Automation, and Web Searching — encompassing 231 tasks. The benchmark evaluates models based on execution outcomes using three evaluator types: format, static, and dynamic evaluators. Experiments across over 20 state-of-the-art proprietary and open-source LLMs reveal that even top models achieve moderate success rates highlighting substantial gaps in long-horizon reasoning, unfamiliar tool usage, and multi-turn context handling. The paper positions MCP-Universe as the first rigorous, extensible testbed for assessing real-world MCP performance and promoting research on robust LLM-agent design.

**Strengths:**

The paper presents a well-motivated and timely contribution that addresses a clear gap in LLM benchmarking — the absence of realistic, execution-based evaluation in real-world MCP contexts. The benchmark design is conceptually coherent and technically grounded: by using authentic servers such as Google Maps, GitHub, and Yahoo Finance, the authors ensure genuine interaction complexity and avoid the artificial constraints of GUI-based or synthetic environments. The inclusion of diverse evaluator types (format, static, dynamic) demonstrates methodological rigor and reduces reliance on subjective “LLM-as-a-judge” paradigms, enhancing reproducibility and fairness. The experimental coverage is broad and systematic, spanning 20+ leading LLMs and multiple agent architectures (ReAct, function calling, enterprise agents), producing a rich quantitative landscape of LLM performance limitations. The paper also provides insightful diagnostic analyses — identifying long-context, unknown-tool, and tool-overload challenges — and introduces preliminary mitigation strategies (e.g., summarization and exploration phases) supported by empirical results. Overall, the work is clearly structured, experimentally thorough, and highly relevant to both academic and industrial AI research on model-agent interoperability.

**Weaknesses:**

the paper lacks statistical robustness i.e. results are presented as raw success rates without standard deviations, error margins, or multiple-run variance, leaving uncertainty about consistency. next, although it identifies critical challenges like long-context failure and tool misuse, the paper does not provide conceptual or theoretical analysis explaining why existing architectures fail — the discussion remains empirical and descriptive rather than explanatory. Similarly, mitigation strategies such as summarization and exploration phases are treated as brief ablations rather than systematically evaluated modules, despite strong quantitative results, the paper misses qualitative depth — it does not include concrete reasoning transcripts or interaction traces showing how model behaviors differ across success and failure cases.

**Questions:**

check weakness

---

> ### Author Response · Authors · 2025-11-26
> **Response to Reviewer ooGm**
>
> Thank you for your insightful reviews! Your professional feedback provides valuable guidance for developing a more comprehensive and competitive paper. We are particularly encouraged that you found our work to be a well-motivated and timely contribution!
>
> > **Weakness**: the paper lacks statistical robustness i.e. results are presented as raw success rates without standard deviations, error margins, or multiple-run variance, leaving uncertainty about consistency.
>
> To address the concern regarding consistency, we conducted a variance analysis by performing 3 independent runs for representative models across different capability tiers. The results, presented in the table below, show low standard deviations, confirming that the benchmark rankings are stable and statistically robust.
>
> | Model  | Overall SR (Mean) | Standard deviation |
> | ---  | ---| --- |
> | GPT-5 | 43.72 | ±0.44 |
> | Claude-4.0-Sonnet | 29.29 | ±1.09 |
> | GLM-4.5 | 24.96 | ±0.90 |
>
> We selected representative models from proprietary SOTA (GPT-5, Claude-4.0-Sonnet) and open-weight SOTA (GLM-4.5) for this analysis. The low variance relative to the performance gaps between models confirms the reliability of our ranking. **We have added this extra analysis in the Appendix E.1 of the paper.**
>
> > **Weakness**: next, although it identifies critical challenges like long-context failure and tool misuse, the paper does not provide conceptual or theoretical analysis explaining why existing architectures fail — the discussion remains empirical and descriptive rather than explanatory.
>
> We performed an **exhaustive manual review** of failure cases from GPT-5 to provide conceptual analysis explaining why existing SOTA LLM fails. As detailed in the table below, we identified five distinct failure mechanisms.
>
> | Failure Category | Percentage |
> |-----------------|------------|
> | Planning Errors | 53.2% |
> | Reasoning Errors | 17.7% |
> | Format Errors | 14.5% |
> | Tool Usage Errors | 8.1% |
> | Context Overflow | 6.5% |
>
> The data highlights that the primary hurdle is Planning Error (53.2%). Models struggle to adapt to runtime obstacles. For instance, in a Booking.com task, the agent encountered a 'flight unavailable' pop-up. Instead of closing the page or modifying the search, it stagnated, repeatedly attempting to scrape the blocked page until it exhausted its step budget. Reasoning failures are also significant; we observed cases where the agent successfully retrieved a price ('SGD 83.00') but inexplicably reported a different number ('SGD 85.00') in the final answer. Additionally, Tool Usage errors reveal that models often miss implicit API constraints, such as setting identical start and end dates in Yahoo Finance, which causes the API to fail. Finally, Context Overflow accounts for failures where the interaction history exceeds the model's maximum context limit. **We have added the error breakdowns in the Appendix E.4 of the paper.**
>
> > **Weakness**: Similarly, mitigation strategies such as summarization and exploration phases are treated as brief ablations rather than systematically evaluated modules, despite strong quantitative results,
>
> We respectfully clarify that these strategies are **not brief ablations but systematic multi-agent and multi-phase workflows** designed to address specific bottlenecks. For summarization, we employ a dual-agent design (a main agent and a summarizer) to compress server outputs at each step. For exploration, we implement a two-phase process: an initial phase to freely explore the MCP server for knowledge acquisition, followed by an exploitation phase. Both designs are directly grounded in the specific challenges identified in our analysis (Long Context and Unknown Tools).
>
> > **Weakness**: the paper misses qualitative depth — it does not include concrete reasoning transcripts or interaction traces showing how model behaviors differ across success and failure cases.
>
> We have addressed this by adding concrete interaction traces in **Appendix E.5**, providing the requested qualitative depth.

---

### Official Review · Reviewer_ciBT · 2025-11-01

**Soundness:** 3
**Presentation:** 3
**Contribution:** 3
**Rating:** 6
**Confidence:** 4

**Summary:**

This paper presents MCP-Universe, a comprehensive benchmark designed to evaluate large language models (LLMs) interacting with real-world tools through the Model Context Protocol (MCP). As MCP rapidly gains adoption across major AI platforms, the benchmark addresses a critical gap in evaluating model performance beyond static text reasoning. It encompasses 231 tasks across six diverse domains such as Location Navigation, Repository Management, Financial Analysis, 3D Design, Browser Automation, and Web Searching using 11 actual MCP servers. The benchmark features a rigorous, execution-based evaluation framework with format, static, and dynamic evaluators that verify results through real API responses rather than subjective judgments. Extensive experiments across more than twenty leading LLMs reveal that even frontier models such as GPT-5-High (44.16% success rate) and Grok-4 (33.33%) struggle considerably, exposing fundamental challenges like long-context reasoning, handling unfamiliar tools, and cross-domain generalization. Overall, the paper is timely, ambitious, and addresses an emerging need in the field of agentic AI evaluation.

The paper provides a highly valuable benchmark that the community will benefit from, especially given the rising importance of MCP-based tool ecosystems. Its engineering quality, clarity, and evaluation rigor are commendable, and it establishes an essential foundation for future research on grounded LLM agents. However, the current version emphasizes system construction and empirical reporting more than analytical or methodological advancement. The experimental results convincingly show that existing models perform poorly, but the paper stops short of offering deep insights or principled pathways for improvement.

In its current form, the work reads as an excellent benchmark and resource paper rather than a full research paper meeting the methodological innovation bar of a top-tier venue. Strengthening the causal analysis of failure modes, expanding the discussion on agent architectures, and providing statistically robust evaluations would significantly elevate its impact. Nonetheless, MCP-Universe is a well-executed and necessary step toward standardized, real-world evaluation of LLM agents, meriting serious consideration and likely strong influence in future benchmarking research.

**Strengths:**

The work’s main strength lies in its timeliness and originality. MCP-Universe is the first benchmark to evaluate LLMs in realistic MCP settings using actual servers rather than simulated environments, making it highly relevant to the evolving AI ecosystem. The benchmark’s breadth and comprehensiveness, spanning multiple domains and including over two hundred tasks demonstrate significant engineering effort and clear understanding of real-world complexity.

The evaluation methodology is robust and thoughtfully designed. By replacing the increasingly common “LLM-as-a-judge” paradigm with execution-based evaluators, the authors achieve greater objectivity and reproducibility. The clear delineation between format, static, and dynamic evaluators ensures rigorous assessment across both static and time-varying tasks.

The paper also excels in clarity and presentation. The problem is well motivated, the related work positioning is thorough, and the figures and tables illustrate the framework and results effectively. The findings especially the gap between model sophistication and real world reliability are clearly articulated and carry high significance for researchers developing tool using agents.

**Weaknesses:**

Despite its strong contribution as a benchmark, the paper’s methodological depth is limited. Its experimental analysis primarily employs standard agent frameworks such as ReAct and basic function calling, offering limited insight into why models fail. While the benchmark surfaces key challenges like long-context reasoning and unfamiliar tool usage, the subsequent analyses of these issues are descriptive rather than diagnostic. The mitigation attempts (summarization and exploration phases) are simple prompt-level strategies with mixed or inconclusive outcomes, lacking principled reasoning or theoretical grounding.

The work would also benefit from deeper failure-mode analysis for example, distinguishing between failures of reasoning, planning, or format compliance and from more detailed justification of certain design choices such as the binary success metric and task difficulty calibration. Additionally, the uneven domain distribution (e.g., heavier weighting toward Web Searching) and absence of human-performance or variance analyses raise questions about benchmark balance and statistical robustness.

These issues do not undermine the benchmark’s value but limit the paper’s standing as a research contribution advancing methodology rather than as a high-quality resource paper.

**Questions:**

Could the authors provide a more detailed breakdown of failure types e.g., incorrect tool selection, parameter misuse, or reasoning lapses and clarify how much each contributes to the overall error rate?


How are dynamic evaluators validated for accuracy when external APIs or data sources change over time?


Have the authors assessed whether some tasks might overlap with training data for large proprietary models, given that MCP servers and examples are public?


Would more advanced agent architectures those incorporating planning, reflection, or persistent memory alter the conclusions about LLM limitations on MCP-Universe?


How might benchmark difficulty be calibrated with respect to human or expert-agent performance to contextualize the reported success rates?

---

> ### Author Response · Authors · 2025-11-26
> **Response to Reviewer ciBT [1/3]**
>
> Thank you for your insightful reviews! Your professional feedback provides valuable guidance for writing a more comprehensive and competitive paper. We are particularly encouraged that you found our work highly valuable and believe that the community will benefit from it.
>
> > **Weakeness 1**: Despite its strong contribution as a benchmark, the paper’s methodological depth is limited.
>
> **First**, we clarify that MCP-Universe is not merely a benchmark but **a comprehensive, modularized evaluation framework** (Section 3). The methodological depth lies in our system (Figure 2), which decouples the agent, MCP environment, and evaluator. This design allows researchers to seamlessly configure and interchange: (1) different LLM backbones (proprietary vs. open-source), (2) distinct agent paradigms (e.g., ReAct vs. Function Calling), (3) diverse real-world MCP servers (e.g., GitHub MCP, Google Maps MCP), and (4) custom evaluation protocols.
>
> **Second**, our work is specifically submitted to the **Datasets and Benchmarks track**, and it aligns strictly with the scope and contributions expected in this area.
>
> > **Weakeness 1**: Its experimental analysis primarily employs standard agent frameworks such as ReAct and basic function calling, offering limited insight into why models fail.
>
> Regarding framework choices, while we employ standard agent frameworks (ReAct, Function Calling) as essential, reproducible baselines for the community, our evaluation extends beyond them. We also include the evaluation of **Enterprise-Level Agents (e.g., Cursor Agent, Claude Code), as detailed in Figure 4**. This allows us to contrast general-purpose baselines against specialized commercial agents.
>
> > **Weakeness 1**: While the benchmark surfaces key challenges like long-context reasoning and unfamiliar tool usage, the subsequent analyses of these issues are descriptive rather than diagnostic.
>
> Our analysis utilizes quantitative data to diagnose specific failures rather than merely describing them:
> - **Long-Context (Figure 5, Left)**: We provide concrete data showing the rapid explosion of token usage as interaction turns increase. This empirically diagnoses that context overflow is one of the challenges.
> - **Unfamiliar-Tools (Figure 6 & Appendix E.5)**: We present detailed case studies and error logs that isolate the model's inability to follow specific MCP tool usages. These examples serve as diagnostic evidence confirming the difficulty of handling unfamiliar tools.
>
> > **Weakeness 1**: The mitigation attempts (summarization and exploration phases) are simple prompt-level strategies with mixed or inconclusive outcomes, lacking principled reasoning or theoretical grounding.
>
> Regarding mitigation strategies, these are not simple prompt-level adjustments but **multi-agent and multi-phase workflows**. For summarization, we employ a dual-agent architecture (a main agent and a summarization agent) to compress server outputs at each step. For exploration, we implement a two-phase process: an initial phase to freely explore the MCP server for knowledge acquisition, followed by an exploitation phase. **Both designs are directly grounded in the specific challenges identified in our analysis (Long Context and Unknown Tools)**.
>
> > **Weakness 2**: The work would also benefit from deeper failure-mode analysis for example, distinguishing between failures of reasoning, planning, or format compliance and from
>
> > **Question 1**: Could the authors provide a more detailed breakdown of failure types e.g., incorrect tool selection, parameter misuse, or reasoning lapses and clarify how much each contributes to the overall error rate?
>
> To address this, we performed an **exhaustive manual review** of failure cases from the SOTA LLM, GPT-5. As detailed in the table below, we identified five distinct failure types.
>
> | Failure Category | Percentage |
> |-----------------|------------|
> | Planning Errors | 53.2% |
> | Reasoning Errors | 17.7% |
> | Format Errors | 14.5% |
> | Tool Usage Errors | 8.1% |
> | Context Overflow | 6.5% |
>
> The data highlights that the primary hurdle is Planning Errors (53.2%). Models struggle to adapt to runtime obstacles. For instance, in a Booking.com task, the agent encountered a 'flight unavailable' pop-up. Instead of closing the page or modifying the search, it stagnated, repeatedly attempting to scrape the blocked page until it exhausted its step budget. Reasoning failures are also significant; we observed cases where the agent successfully retrieved a price ('SGD 83.00') but inexplicably reported a different number ('SGD 85.00') in the final answer. Additionally, Tool Usage errors reveal that models often miss implicit API constraints, such as setting identical start and end dates in Yahoo Finance, which causes the API to fail. Finally, Context Overflow accounts for failures where the interaction history exceeds the model's maximum context limit. **A discussion of these error breakdowns is added in Appendix E.4.**

---

> ### Author Response · Authors · 2025-11-26
> **Response to Reviewer ciBT [2/3]**
>
> > **Weakness 2**: more detailed justification of certain design choices such as the binary success metric and task difficulty calibration.
>
> Regarding Evaluation Metrics, we employ a complementary dual-metric system: Average Evaluator Score (AE) and Success Rate (SR).
> - Following OSWorld, the Average Evaluator Score represents the mean percentage of passed evaluators per task. This ensures models receive partial credit for progress, preventing a zero score due to a single missed evaluator.
> - The Success Rate serves as a strict binary metric to determine if the task was fully completed.
>
> These metrics work together to provide a balanced evaluation: the AE rewards progress, while the SR ensures that high partial scores do not mask the inability to fully complete the task. **We have added a detailed discussion of these metrics to the final paragraph of Section 3.4.**
>
> Regarding Task Difficulty Calibration, we implemented a rigorous filtering protocol. Tasks that could be trivially solved by LLMs without MCP servers, or consistently completed within five retries using MCP servers, were discarded and replaced to ensure the benchmark remains challenging.
>
> > **Werakness 2**: Additionally, the uneven domain distribution (e.g., heavier weighting toward Web Searching) and absence of human-performance or variance analyses raise questions about benchmark balance and statistical robustness.
>
> > **Question 5**: How might benchmark difficulty be calibrated with respect to human or expert-agent performance to contextualize the reported success rates?
>
> Regarding domain distribution, we respectfully clarify that our benchmark is balanced. Figure 3 illustrates our coverage across 6 primary domains, which are further granularized into 27 sub-domains. Notably, no single sub-domain exceeds 9% of the total tasks. This data confirms that the benchmark maintains a diverse and balanced distribution, rather than being heavily weighted toward any single task type.
>
> Regarding Human Performance, we note that direct interaction with raw MCP servers is not a natural way for humans. However, to calibrate difficulty and ensure task feasibility, we invited human evaluators to assess these tasks using standard Computer GUIs. The results reveal that **all tasks** are solvable by humans. This confirms that the lower success rates reported for models reflect agentic limitations rather than task infeasibility.
>
> > **Weakness 3**: These issues do not undermine the benchmark’s value but limit the paper’s standing as a research contribution advancing methodology rather than as a high-quality resource paper.
>
> We appreciate the reviewer's recognition of the benchmark's value. We would like to clarify that this work is submitted to the **Datasets and Benchmarks area**. As such, our primary contribution is intended to be a high-quality resource and rigorous evaluation standard for the community.
>
> > **Question 2**: How are dynamic evaluators validated for accuracy when external APIs or data sources change over time?
>
> **Regarding API Stability**, we rely on official, enterprise-level MCP servers wrapping well-established APIs (e.g., Google Maps, GitHub). These services are highly stable and maintained for long-term backward compatibility.
>
> **Regarding Data Dynamics**, we ensure accuracy by fixing the query parameters while retrieving ground truth dynamically. For example, in a weather-related task, the target location is fixed in the dataset, but the evaluator queries the live API in real-time to obtain current data. This allows the evaluator to validate the model's output against the actual state of the world at the specific moment of inference.
>
> > **Question 3**: Have the authors assessed whether some tasks might overlap with training data for large proprietary models, given that MCP servers and examples are public?
>
> We confirm that overlap with training data is **highly unlikely**. All tasks in MCP-Universe were newly created specifically for this benchmark, ensuring that these specific task instances do not exist in the training corpora of current models.

---

> ### Author Response · Authors · 2025-11-26
> **Response to Reviewer ciBT [3/3]**
>
> > **Question 4**: Would more advanced agent architectures those incorporating planning, reflection, or persistent memory alter the conclusions about LLM limitations on MCP-Universe?
>
> Yes, the incorporation of planning, reflection, and persistent memory yields performance gains. We validate this empirically in **Appendix H** by introducing HarmonyReAct, an agent framework explicitly designed to integrate these capabilities and mitigate model limitations:
> - **Planning**: We enforce a dedicated analysis channel before every action, compelling the model to perform explicit task decomposition and planning.
> - **Reflection**: We implement a Self-Correction Mechanism. If the agent generates invalid output, the system feeds back specific error signals (e.g., format violations), triggering a reflection loop where the model analyzes and corrects its previous error.
> - **Memory Management**: We utilize a Summarize-and-Wipe strategy. Instead of simple truncation, the agent actively summarizes verbose tool outputs.
>
> **Experimental Evidence**: This agent framework yields substantial gains:
> - GPT-OSS-120B (Native Function Call): 25.54% Success Rate
> - HarmonyReAct (GPT-OSS-120B): 31.17% Success Rate (matching Claude-4.0-Sonnet)

---

### Author Response · Authors · 2025-12-01
**Summary of Our Rebuttal to the Area Chair**

To the Area Chair,

We are writing to summarize our rebuttal process and the consensus reached before the reviewer information leak.
Due to the rollback, the currently visible scores are 6, 6, 6, and 4. However, **before the information leak, Reviewer Wvi6 (initially a 4) explicitly confirmed that their concerns were resolved and raised their score to 6, so the overall score should be 6, 6, 6, 6.**

**1. Addressed Concerns and Rating Update (Reviewer Wvi6)**

Reviewer Wvi6 initially gave a score of 4, citing a lack of systematic error analysis and cost metrics. In our response, we provided:
- Systematic Error Analysis: A detailed manual breakdown of failure types (e.g., Planning Errors 53.2%, Reasoning 17.7%).
- Cost & Efficiency Metrics: Detailed token usage and API cost tables for SOTA models.
- Interaction Traces: Qualitative examples of "unknown tool" failures.

All of them have been included in the revised paper.

**Outcome**: On Nov 26, Reviewer Wvi6 responded:

"Thanks for the detailed response. My primary issues are resolved; I have raised my score."

**The score was raised to 6 on Nov. 26, before the date of the information leak.**

**2. Addressed Common Concerns**

**All reviewers (ciBT, ooGm, eTRd, Wvi6) acknowledged the timeliness and strong quality of the benchmark**. We addressed their shared concerns as follows:

- Failure Analysis (Addressed ciBT, ooGm, Wvi6, eTRd): We moved beyond simple success rates by conducting an exhaustive manual review of GPT-5 failure cases. We added a new section/appendix detailing that Planning (53.2%) is the primary bottleneck, followed by Reasoning and Format errors. This provides the diagnostic depth requested by all reviewers.
- Statistical Robustness & Metrics (Addressed ooGm, eTRd, Wvi6):
  - Variance: We added a variance analysis (3 independent runs) showing low standard deviation, confirming ranking stability.
  - Cost/Efficiency: We added a comprehensive cost analysis (tokens/USD) for evaluating the full benchmark.
  - Scoring: We clarified the dual-metric system (Success Rate vs. Average Evaluator Score) to explain "last-mile failures" where models perform well but fail final strict evaluators.
- Advanced Agent Architectures (Addressed ciBT, eTRd): To prove the benchmark’s utility for research, we introduced HarmonyReAct (Appendix H), an agent incorporating planning, reflection, and memory. This architecture improved success rates significantly (from 25.5% to 31.2%), demonstrating that the benchmark effectively measures architectural improvements.

**3. Conclusion**

MCP-Universe is the first benchmark to evaluate LLMs using real-world MCP servers. **We have addressed all reviewer questions**, improving clarity, and have added both qualitative and quantitative analyses. **All new results and analyses are included in the revised paper.**

Thank you for your time and consideration during this complex review cycle.

---

### Meta-Review · Area_Chair_3aY5 · 2026-01-07

**Summary:**

This paper presents a timely resource for evaluating LLM-based agents using real-world MCP servers, which constitutes its primary contribution. The benchmark provides a useful dataset and initial empirical findings, and the experiments reveal meaningful performance trends across models and tasks.

However, the paper also exhibits several limitations that constrain its methodological contribution. While a high-quality benchmark paper should not only provide well-curated resources but also offer deeper diagnostic insights into why different models fail, the current analysis remains largely descriptive. The failure-mode analysis is incomplete and does not sufficiently distinguish between different types of errors. In addition, the benchmark suffers from uneven domain coverage, task-selection biases, and limited consideration of efficiency-related metrics. Furthermore, the experimental evaluation focuses on a narrow set of agent frameworks, without broader comparisons across alternative agent architectures.

I tend to reject this paper.

**Reviewer Concerns:**

I believe that, as a resource paper, this timely benchmark undoubtedly makes a valuable contribution to the community. However, the paper still lacks deeper analytical insights, and the current data analysis remains largely descriptive rather than diagnostic. That said, in their rebuttal, the authors partially addressed the reviewers’ concerns by adding more systematic error analysis and extending the coverage of evaluation metrics.

**Reviewer Scores:**

Reviewer ciBT: This reviewer believes that the rebuttal does not adequately address the concern regarding the lack of deep insight into failure cases and the underlying reasons why models fail. As a result, the reviewer is likely to maintain a score of 6.

Reviewer ooGm: This reviewer raised concerns about statistical robustness and the lack of qualitative depth. Based on the authors’ rebuttal, these concerns were not sufficiently addressed, and therefore the reviewer is likely to maintain a score of 6.

Reviewer Wvi6: The main concerns were related to error analysis and metric coverage. These issues were addressed in the rebuttal, and as a result, the reviewer is likely to raise their score from 4 to 6.

Reviewer eTRd: This reviewer questioned the lack of diversity in the underlying problem structures within each domain. The authors’ response did not sufficiently resolve this concern, and the reviewer is likely to maintain their original score 6.

---

### Decision · Program_Chairs · 2026-01-26

Reject